# RB-Modulation: Training-Free Stylization using Reference-based Modulation

**Litu Rout**[1,2*]   **Yujia Chen**[1]   **Nataniel Ruiz**[1]
**Abhishek Kumar**[3]   **Constantine Caramanis**[2]   **Sanjay Shakkottai**[2]   **Wen-Sheng Chu**[1]
[1] Google   [2] UT Austin   [3] Google DeepMind
{litu.rout,constantine,sanjay.shakkottai}@utexas.edu
{liturout,yujiachen,natanielruiz,abhishk,wschu}@google.com

## ABSTRACT

We propose Reference-Based Modulation (RB-Modulation), a new plug-and-play solution for training-free personalization of diffusion models. Existing training-free approaches exhibit difficulties in (a) style extraction from reference images in the absence of additional style or content text descriptions, (b) unwanted content leakage from reference style images, and (c) effective composition of style and content. RB-Modulation is built on a novel stochastic optimal controller where a style descriptor encodes the desired attributes through a terminal cost. The resulting drift not only overcomes the difficulties above, but also ensures high fidelity to the reference style and adheres to the given text prompt. We also introduce a cross-attention-based feature aggregation scheme that allows RB-Modulation to decouple content and style from the reference image. With theoretical justification and empirical evidence, our test-time optimization framework demonstrates precise extraction and control of *content* and *style* in a training-free manner. Further, our method allows a seamless composition of content and style, which marks a departure from the dependency on external adapters or ControlNets. See project page https://rb-modulation.github.io/ for code and further details.

## 1 INTRODUCTION

Text-to-image (T2I) generative models (Ramesh et al., 2021; Rombach et al., 2022; Saharia et al., 2022) have excelled in crafting visually appealing images from text prompts. These T2I models are increasingly employed in creative endeavors such as visual arts (Xu et al., 2024), gaming (Pearce et al., 2023), personalized image synthesis (Ruiz et al., 2023; Huang et al., 2024a; Hu et al., 2021; Shah et al., 2023), stylized rendering (Sohn et al., 2023; Hertz et al., 2023; Wang et al., 2024a; Jeong et al., 2024), and image inversion or editing (Ulyanov et al., 2018; Delbracio & Milanfar, 2023; Rout et al., 2023b; 2024; Mokady et al., 2023). Content creators often need precise control over both the *content* and the *style* of generated images to match their vision. While the content of an image can be conveyed through text, articulating an artist's unique style – characterized by distinct brushstrokes, color palette, material, and texture – is substantially more nuanced. This has led to research on personalization through visual prompting (Sohn et al., 2023; Hertz et al., 2023; Wang et al., 2024a).

Recent studies have focused on finetuning pre-trained T2I models to learn style from a set of reference images (Gal et al., 2022; Ruiz et al., 2023; Sohn et al., 2023; Hu et al., 2021). This involves optimizing the model's text embeddings, model weights, or both, using the denoising diffusion loss. However, these methods demand substantial computational resources for training or finetuning large-scale foundation models, thus making them expensive to adapt to new, unseen styles. Furthermore, these methods often depend on human-curated images of the same style, which is less practical and can compromise quality when only a single reference image is available.

In training-free **stylization**, recent methods (Hertz et al., 2023; Wang et al., 2024a; Jeong et al., 2024) manipulate keys and values within the attention layers using just one reference style image. These methods face challenges in both extracting the style from the reference style image and accurately transferring the style to a target content image. For instance, during the DDIM inversion step (Song et al., 2021a) utilized by StyleAligned (Hertz et al., 2023), fine-grained details tend to be compromised. To mitigate this issue, InstantStyle (Wang et al., 2024a) incorporates features from

---

*This work was done during an internship at Google.

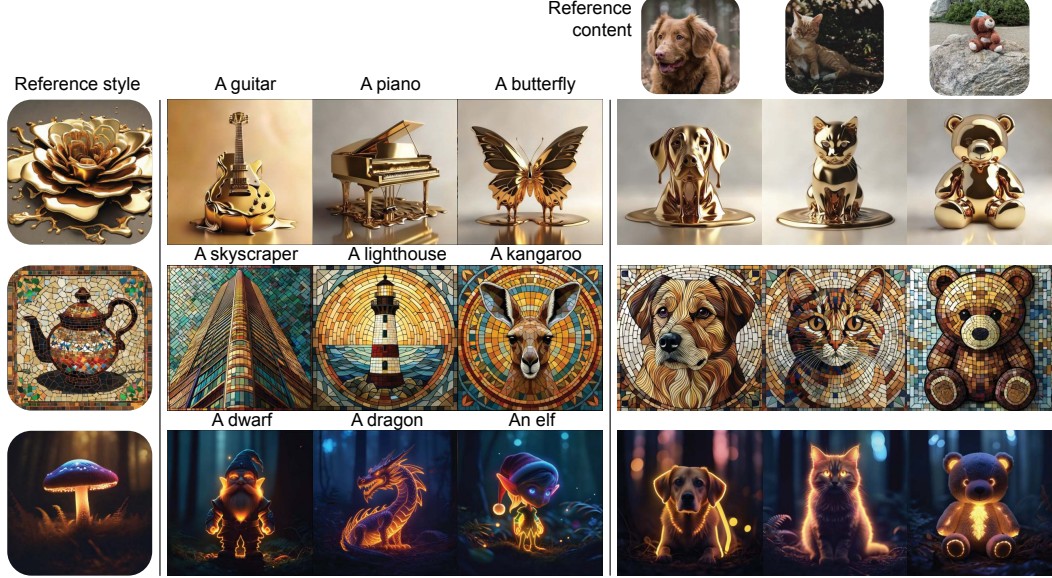

Figure 1: Given a single reference image (rounded rectangle), our method **RB-Modulation** offers a plug-and-play solution for (a) stylization, and (b) content-style composition with various prompts while maintaining sample diversity and prompt alignment. For instance, given a reference style image (e.g., "melting golden 3d rendering style") and content image (e.g., "a dog"), our method adheres to the desired prompts without leaking contents (e.g., flower) from the reference style image and without being restricted to the fixed pose or layout of the reference dog image.

the reference style image into specific layers of a previously trained IP-Adapter (Ye et al., 2023). However, identifying the exact layer for feature injection in a model is complex and not universally applicable across models. Also, feature injection can cause content leakage from the style image into the generated content. Moving on to content-style **composition**, InstantStyle (Wang et al., 2024a) employs a ControlNet (Zhang et al., 2023) (an additionally trained network) to preserve image layout, which inadvertently limits its diversity.

We introduce Reference-Based Modulation (RB-Modulation), a novel approach for stylization and composition that eliminates the need for training or finetuning diffusion models (*e.g.* Control-Net (Zhang et al., 2023) or adapters (Ye et al., 2023; Hu et al., 2021)). Our work reveals that the reverse dynamics in diffusion models can be formulated as stochastic optimal control problem. By incorporating style features into the controller's terminal cost, we modulate the drift field in diffusion model's reverse dynamics, enabling training-free personalization. Unlike conventional attention processors that often leak content from the reference style image, we propose to enhance the image fidelity via an Attention Feature Aggregation (AFA) module that decouples content from reference style image. We demonstrate the effectiveness of our method in stylization (Hertz et al., 2023; Wang et al., 2024a; Jeong et al., 2024) and style+content composition, as illustrated in Figure 1(a) and (b), respectively. Our experiments show that RB-Modulation outperforms current SoTA methods (Hertz et al., 2023; Wang et al., 2024a) in terms of human preference and prompt-alignment metrics.

**Our contributions are summarized as follows:**

- We present reference-based modulation (RB-Modulation), a novel stochastic optimal control based test-time optimization framework that enables training-free, personalized style and content control, with a new Attention Feature Aggregation (AFA) module to maintain high fidelity to the reference image while adhering to the given prompt (§4).
- We provide theoretical justifications connecting optimal control and reverse diffusion dynamics. We leverage this connection to incorporate desired attributes (*e.g.*, style) in our controller's terminal cost and personalize T2I models in a training-free manner (§5).
- We perform extensive experiments covering stylization and content-style composition, demonstrating superior performance over SoTA methods in human preference metrics (§6).

## 2 RELATED WORK

**Personalization of T2I models:** T2I generative models (Rombach et al., 2022; Podell et al., 2023; Pernias et al., 2024) can now generate high quality images from text prompts. Their text-following

ability has unlocked new avenues in personalized content creation, including text-guided image editing (Mokady et al., 2023; Rout et al., 2024), solving inverse problems (Rout et al., 2023b; 2024), concept-driven generation (Ruiz et al., 2023; Tewel et al., 2023; Kumari et al., 2023; Chen et al., 2024), personalized outpainting (Tang et al., 2023), identity-preservation (Ruiz et al., 2024; Huang et al., 2024a; Wang et al., 2024b), and stylized synthesis (Sohn et al., 2023; Wang et al., 2024a; Hertz et al., 2023; Shah et al., 2023). To tailor T2I models for a specific style (*e.g.*, painting) or content (*e.g.*, object), existing methods follow one of two recipes: (1) full finetuning (FT) or parameter efficient finetuning (PEFT) and (2) training-free, which we discuss below.

**Finetuning T2I models for personalization:** FT (Ruiz et al., 2023; Everaert et al., 2023) and PEFT (Kumari et al., 2023; Hu et al., 2021; Sohn et al., 2023; Shah et al., 2023) methods excel at capturing style or object details when the underlying T2I model can be finetuned on a few (typically 4) reference images for few thousand iterations. PARASOL (Tarrés et al., 2024) requires supervised data via a cross-modal search to train both the denoising U-Net and a projector network. Diff-NST (Ruta et al., 2023) trains the attention processor by targeting the 'V' values within the denoising U-Net. The curation of supervised data and resource-intensive finetuning for every style or content makes these methods challenging for practical usage.

**Training-free methods for personalization:** Training-free personalization methods are preferable to finetuning methods given the vastly faster time of execution. In **StyleAligned** (Hertz et al., 2023), a reference style image and a text prompt describing the style are used to extract style features via DDIM inversion (Song et al., 2021a). Target queries and keys are then normalized using adaptive instance normalization (Huang & Belongie, 2017) based on reference counterparts. Finally, reference image keys and values are merged with DDIM-inverted latents in self-attention layers, which tends to leak content information from the reference style image (Figure 2). Moreover, the need for textual description in the DDIM inversion step can degrade its performance. **DiffusionDisentanglement** (Wu et al., 2023) aims to reduce the approximation error in DDIM inversion by jointly minimizing a perceptual loss and a directional CLIP loss, which is prone to content leakage (Wang et al., 2024a). **Swapping Self-Attention (SSA)** (Jeong et al., 2024) addresses these limitations by replacing the target keys and values in self-attention layers with those from a reference style image. It still relies on DDIM inversion to cache keys and values of the reference style, which tends to compromise fine-grained details (Wang et al., 2024a). Both StyleAligned (Hertz et al., 2023) and SSA (Jeong et al., 2024) require two reverse processes to share their attention layer features and thus demand significant memory. **InstantStyle** (Wang et al., 2024a) injects reference style features into specific cross-attention layers of IP-Adapter (Ye et al., 2023), addressing two key limitations: DDIM inversion and memory-intensive reverse processes. However, pinpointing the exact layers for feature injection is complex, and may not generalize to other models. In addition, when composing style and content, InstantStyle (Wang et al., 2024a) relies on ControlNet (Zhang et al., 2023), which can limit the diversity of generated images to fixed layouts and deviate from the prompt.

**Optimal Control:** Stochastic optimal control finds wide applications in diverse fields such as molecular dynamics (Holdijk et al., 2024), economics (Fleming & Rishel, 2012), non-convex optimization (Chaudhari et al., 2018), robotics (Theodorou et al., 2011), and mean-field games (Carmona et al., 2018) Despite its extensive use, and recent works on its connections to diffusion based generative models (Berner et al., 2024; Tzen & Raginsky, 2019; Chen et al., 2023), it has been less explored in training-free personalization. In this paper, we introduce a novel test-time optimization framework leveraging the main concepts from optimal control to achieve training-free personalization. A key aspect of optimal control is designing a controller to guide a stochastic process towards a desired terminal condition (Fleming & Rishel, 2012). This aligns with our goal of training-free personalization, as we target a specific style or content at the end of the reverse diffusion process, which can be incorporated in the controller's terminal condition.

RB-Modulation overcomes several challenges encountered by SoTA methods (Hertz et al., 2023; Jeong et al., 2024; Wang et al., 2024a). Since RB-Modulation does not require DDIM inversion, it retains fine-grained details unlike StyleAligned (Hertz et al., 2023). Using a stochastic controller to refine the trajectory of a single reverse process, it overcomes the limitation of coupled reverse processes (Hertz et al., 2023). By incorporating a style descriptor in our controller's terminal cost, it eliminates the dependency on Adapters (Ye et al., 2023; Hu et al., 2021) or ControlNets (Zhang et al., 2023) by InstantStyle (Wang et al., 2024a).

## 3 PRELIMINARIES

**Diffusion models** consist of two stochastic processes: (a) *noising process*, modeled by a Stochastic Differential Equation (SDE) known as forward-SDE: $dX_t = f(X_t, t) \, dt + g(X_t, t) \, dW_t, X_0 \sim p_0$, and (b) *denoising process*, modeled by the time-reversal of forward-SDE under mild regularity conditions (Anderson, 1982), also known as reverse-SDE:

$$dX_t = \left[ f(X_t, t) - g^2(X_t, t) \nabla \log p(X_t, t) \right] dt + g(X_t, t) \, dW_t, \qquad X_1 \sim \mathcal{N}(0, I_d). \quad (1)$$

Here, $W = (W_t)_{t \geq 0}$ is standard Brownian motion in a filtered probability space, $(\Omega, \mathcal{F}, (\mathcal{F}_t)_{t \geq 0}, \mathcal{P})$, $p(\cdot, t)$ denotes the marginal density of $p$ at time $t$, and $\nabla \log p_t(\cdot)$ the corresponding score function. $f(X_t, t)$ and $g(X_t, t)$ are called drift and volatility, respectively. A popular choice of $f(X_t, t) = -X_t$ and $g(X_t, t) = \sqrt{2}$ corresponds to the well-known forward Ornstein-Uhlenbeck (OU) process.

For T2I generation, the reverse-SDE (1) is simulated using a neural network $s(\mathbf{x}_t, t; \theta)$ (Hyvärinen & Dayan, 2005; Vincent, 2011) to approximate $\nabla_{\mathbf{x}} \log p(\mathbf{x}_t, t)$. Importantly, to accelerate the sampling process in practice (Song et al., 2021a; Karras et al., 2022; Zhang & Chen, 2022), the reverse-SDE (1) shares the same path measure with a probability flow ODE: $dX_t = \left[ f(X_t, t) - \frac{1}{2} g^2(X_t, t) \nabla \log p(X_t, t) \right] dt$, where $X_1 \sim \mathcal{N}(0, I_d)$.

**Personalized diffusion models** either fully finetune $\theta$ of $s(\mathbf{x}_t, t; \theta)$ (Ruiz et al., 2023; Everaert et al., 2023), or train a parameter-efficient adapter $\Delta\theta$ for $s(\mathbf{x}_t, t; \theta + \Delta\theta)$ on reference style images (Hu et al., 2021; Sohn et al., 2023; Shah et al., 2023). Our method does not finetune $\theta$ or train $\Delta\theta$. Instead, we derive a new drift field through a stochastic control that *modulates* the reverse-SDE (1).

## 4 METHOD

**Personalization using optimal control:** Normalize time $t$ by the total number of diffusion steps $T$ such that $0 \leq t \leq 1$. Let us denote by $u : \mathbb{R}^d \times [0, 1] \to \mathbb{R}^d$ a controller from the admissible set of controls $\mathcal{U} \subseteq \mathbb{R}^d$, $X_t^u \in \mathbb{R}^d$ a state variable, $\ell : \mathbb{R}^d \times \mathbb{R}^d \times [0, 1] \to \mathbb{R}$ the transient cost, and $h : \mathbb{R}^d \to \mathbb{R}$ the terminal cost of the reverse process $(X_t^u)_{t=1}^0$. We show in §5 that training-free personalization can be formulated as a control problem where the drift of the standard reverse-SDE (1) is modified via RB-modulation:

$$\min_{u \in \mathcal{U}} \mathbb{E}\left[ \int_1^0 \ell(X_t^u, u(X_t^u, t), t) \, dt + \gamma h(X_0^u) \right], \quad \text{where} \quad (2)$$
$$dX_t^u = \left[ f(X_t^u, t) - g^2(X_t^u, t) \nabla \log p(X_t^u, t) + u(X_t^u, t) \right] dt + g(X_t^u, t) dW_t, X_1^u \sim \mathcal{N}(0, I_d).$$

Importantly, the terminal cost $h(\cdot)$, weighted by $\gamma$, captures the discrepancy in feature space between the styles of the reference image and the generated image. The resulting controller $u(\cdot, t)$ modulates the drift over time to satisfy this terminal cost. We derive the solution to this optimal control problem through the Hamilton-Jacobi-Bellman (HJB) equation (Fleming & Rishel, 2012); refer to Appendix A for details. Our proposed RB-Modulation **Algorithm 1** has two key components: (a) stochastic optimal controller and (b) attention feature aggregation. Below, we discuss each in turn.

**(a) Stochastic Optimal Controller (SOC):** We show that the reverse dynamics in diffusion models can be framed as a stochastic optimal control problem with a quadratic terminal cost (theoretical analysis in §5). For personalization using a reference style image $X_0^f = \mathbf{z}_0$, we use a Contrastive Style Descriptor (CSD) (Somepalli et al., 2024) to extract style features $\Psi(X_0^f)$. Since the score functions $s(\mathbf{x}_t, t; \theta) \approx \nabla \log p(X_t, t)$ are available from pre-trained diffusion models (Podell et al., 2023; Pernias et al., 2024), our goal is to add a correction term $u(\cdot, t)$ to modulate the reverse-SDE and minimize the overall cost (2). We approximate $X_0^u$ with its conditional expectation using Tweedie's formula (Efron, 2011; Rout et al., 2023b; 2024). Finally, we incorporate the style features into our controller's terminal cost as: $h(X_0^u) = \|\Psi(X_0^f) - \Psi(\mathbb{E}[X_0^u | X_t^u])\|_2^2$.

Our theoretical results (§5) suggest that the optimal controller can be obtained by solving the HJB equation and letting $\gamma \to \infty$. In practice, this translates to dropping the transient cost $\ell(X_t^u, u(X_t^u, t), t)$ and solving (2) with only the terminal constraint, *i.e.*,

$$\min_{u \in \mathcal{U}} \|\Psi(X_0^f) - \Psi(\mathbb{E}[X_0^u | X_t^u])\|_2^2. \quad (3)$$

Thus, we solve (3) to find the optimal control $u$ and use this controller in the reverse dynamics (2) to update the current state from $X_t^u$ to $X_{t-\Delta t}^u$ (recall that time flows backwards in the reverse-SDE (1)). Our implementation of (3) is given in **Algorithm 1**, which follows from our theoretical insights.

**Implementation challenge:** For smaller models (Rombach et al., 2022), we can directly solve our control problem (3). However, for larger models (Podell et al., 2023; Pernias et al., 2024), the control objective (3) requires back propagation through the score network with tentatively billions of parameters. This significantly increases time and memory complexity (Rout et al., 2023b; 2024).

We propose a test-time proximal gradient descent approach to address this challenge. The key ingredient of our **Algorithm 1** is to find the previous state $X_{t-\Delta t}$ by modulating the current state $X_t$ based on an optimal controller $u^*$. The optimal controller $u^*$ is obtained by minimizing the discrepancy in style between $\bar{X}_0^u := \mathbb{E}[X_0^u | X_t^u = \mathbf{x}_t]$, obtained using our controlled reverse-SDE (3), and the reference style image $\mathbf{z}_0$. Motivated by this interpretation, an alternate **Algorithm 2** avoids back propagation through $s(\mathbf{x}_t, t; \theta)$ by introducing a dummy variable $\mathbf{x}_0$, which serves as a proxy for $\bar{X}_0^u$ in the terminal cost. Instead of forcing $\mathbf{x}_0$ to be decided by the dynamics of the reverse-SDE as in **Algorithm 1**, we allow it to be only approximately faithful to the dynamics. This is implemented by adding a proximal penalty, *i.e.* $\mathbf{x}_0^* = \arg\min_{\mathbf{x}_0 \in \mathbb{R}^d} \|\Psi(X_0^f) - \Psi(\mathbf{x}_0)\|_2^2 + \lambda \|\mathbf{x}_0 - \mathbb{E}[X_0^u | X_t^u = \mathbf{x}_t]\|_2^2$, where the hyper-parameter $\lambda$ controls the faithfulness of the reverse dynamics. This penalty assumes that with a small step-size in (3), $\mathbf{x}_0^*$ and $\mathbb{E}[X_0^u | X_t^u = \mathbf{x}_t]$ will be close. Thus, **Algorithm 2** enables personalization of large-scale foundation models, *matching the speed of training-free methods and obtaining 5-20X speedup over training-based methods*; see Table 4 in Appendix B.2 for details.

While prior works (Chung et al., 2023; Zhu et al., 2023; He et al., 2024) have used a proximal sampler in related settings, their underlying generative model is not personalized. We believe that this is an important reason why our method results in a significant speedup while satisfying the terminal constraints. Our paper takes the first step in personalizing the underlying generative model via a novel attention processor as discussed below.

**(b) Attention Feature Aggregation (AFA):** Let $d$ denote the dimension of the latent variable $X_t$, $n_q$ the embedding dimension for query $Q$, and $n_h$ the output dimension of the hidden layer. Transformer-based diffusion models (Rombach et al., 2022; Podell et al., 2023; Pernias et al., 2024) consist of self-attention and cross-attention layers operating on latent embedding $\mathbf{x}_t \in \mathbb{R}^{d \times n_h}$. Within the attention module $\text{Attention}(Q, K, V)$, $\mathbf{x}_t$ is projected into queries $Q \in \mathbb{R}^{d \times n_q}$, keys $K \in \mathbb{R}^{d \times n_q}$, and values $V \in \mathbb{R}^{d \times n_h}$ using linear projections. Through $Q$, $K$, and $V$, attention layers capture global context and improve long-range dependencies within $\mathbf{x}_t$.

To incorporate a reference image (*e.g.*, style or content) while retaining alignment with the prompt, we introduce the Attention Feature Aggregation (AFA) module. Given a prompt $\mathbf{p}$, a reference style image $I_s$, and a reference content image $I_c$, we first extract the embeddings using CLIP text encoder (Radford et al., 2021) and CSD image encoder (Somepalli et al., 2024). These embeddings are projected into keys and values using linear projection. We denote by $K_p$ and $V_p$ the keys and values from $\mathbf{p}$, $K_s$ and $V_s$ from $I_s$, $K_c$ and $V_c$ from $I_c$ (used only in content-style composition). The query $Q$, derived from a linear projection of $\mathbf{x}_t$, remains consistent in the AFA module. To maintain consistency between text and style, we compose the keys and values of both text and style in our attention mechanism. The final output of the AFA module is given by

$$AFA = \text{Avg}\left(A_{text}, A_{style}, A_{text+style}\right), A_{text} = \text{Attention}(Q, [K; K_p], [V; V_p]),$$
$$A_{style} = \text{Attention}(Q, [K; K_s], [V; V_s]), A_{text+style} = \text{Attention}(Q, [K; K_p; K_s], [V; V_p; V_s]),$$

where $[K; K_p] \in \mathbb{R}^{2d \times n_q}$ indicates concatenation of $K$ with $K_p$ along the number of tokens dimension. For style-content composition, we process the content image $I_c$ in the same way as the reference style image $I_s$, and obtain another set of attention outputs:

$$AFA = \text{Avg}\left(A_{text}, A_{style}, A_{content}, A_{content+style}\right),$$
$$A_{content} = \text{Attention}(Q, [K; K_c], [V; V_c]), A_{content+style} = \text{Attention}(Q, [K; K_s; K_c], [V; V_s; V_c]).$$

Importantly, the AFA module is computationally tractable as it only requires the computation of a multi-head attention, which is widely used in practice (Podell et al., 2023).

**Disentangling content and style.** In stylization (content described by text; style illustrated by a reference style image), prior works (Hertz et al., 2023; Wang et al., 2024a) inject the entire reference style image $I_s$ that does not disentangle content and style. However, our AFA module injects

**Algorithm 1:** RB-Modulation (Exact)

**Input:** Diffusion steps $T$, reference prompt $\mathbf{p}$, reference style image $\mathbf{z}_0$, style descriptor $\Psi(\cdot)$, score network $s(\cdot, \cdot, \cdot; \theta)$

**Tunable parameter:** Stepsize $\eta$, optimization steps $M$

**Output:** Personalized latent $X_0^u$

1   Initialize $\mathbf{x}_T \leftarrow \mathcal{N}(0, \mathrm{I}_d)$
2   **for** $t = T$ **to** $1$ **do**
3     Initialize controller $u = 0$
4     **for** $m = 1$ **to** $M$ **do**
5       $\hat{\mathbf{x}}_t = \mathbf{x}_t + u$     ▷ controlled state
6       $\bar{X}_0^u = \frac{\hat{\mathbf{x}}_t}{\sqrt{\bar{\alpha}_t}} + \frac{(1 - \bar{\alpha}_t)}{\sqrt{\bar{\alpha}_t}} s(\hat{\mathbf{x}}_t, t, \mathbf{p}; \theta)$
7       $h(\bar{X}_0^u) = \|\Psi(\mathbf{z}_0) - \Psi(\bar{X}_0^u)\|_2^2$ using Eq. (3)
8       $u = u - \eta \nabla_u h(\bar{X}_0^u)$    ▷ update controller
9     **end**
10    $\mathbf{x}_t^* = \mathbf{x}_t + u$     ▷ optimally controlled state
11    $\bar{X}_0^u = \frac{\mathbf{x}_t^*}{\sqrt{\bar{\alpha}_t}} + \frac{(1 - \bar{\alpha}_t)}{\sqrt{\bar{\alpha}_t}} s(\mathbf{x}_t^*, t, \mathbf{p}; \theta)$ ▷ terminal state
12    $\mathbf{x}_{t-1} \leftarrow \mathrm{DDIM}(\bar{X}_0^u, \mathbf{x}_t^*)$     ▷ one denoising update
13   **end**
14   **return** $X_0^u$

**Algorithm 2:** RB-Modulation (Proximal)

**Input:** Diffusion time steps $T$, reference prompt $\mathbf{p}$, reference style image $\mathbf{z}_0$, style descriptor $\Psi(\cdot)$, score network $s(\cdot, \cdot, \cdot; \theta)$

**Tunable parameters:** Stepsize $\eta$, optimization steps $M$, proximal strength $\lambda$

**Output:** Personalized latent $X_0^u$

1   Initialize $\mathbf{x}_T \leftarrow \mathcal{N}(0, \mathrm{I}_d)$
2   **for** $t = T$ **to** $1$ **do**
3     Compute posterior mean
      $\mathbb{E}[X_0^u | X_t^u = \mathbf{x}_t] = \frac{\mathbf{x}_t}{\sqrt{\bar{\alpha}_t}} + \frac{(1 - \bar{\alpha}_t)}{\sqrt{\bar{\alpha}_t}} s(\mathbf{x}_t, t, \mathbf{p}; \theta)$
4     Initialize opt. variable $\mathbf{x}_0 = \mathbb{E}[X_0^u | X_t^u = \mathbf{x}_t]$
5     **for** $m = 1$ **to** $M$ **do**
6       Compute controller's cost $\mathcal{L}(\mathbf{x}_0) := \|\Psi(\mathbf{z}_0) - \Psi(\mathbf{x}_0)\|_2^2 + \lambda \|\mathbf{x}_0 - \mathbb{E}[X_0^u | X_t^u = \mathbf{x}_t]\|_2^2$
7       Update optimization variable
       $\mathbf{x}_0 = \mathbf{x}_0 - \eta \nabla_{\mathbf{x}_0} \mathcal{L}(\mathbf{x}_0)$
8     **end**
9     $\mathbf{x}_{t-1} \leftarrow \mathrm{DDIM}(\mathbf{x}_0, \mathbf{x}_t)$     ▷ one denoising step
10   **end**
11   **return** $X_0^u$

*only the style features* from $I_s$ using the style attention head of the Vision Transformer (ViT) in CSD (Somepalli et al., 2024). The AFA module achieves content-style disentanglement by computing separate attention maps for content from text and style from image. In this case, SOC does not handle content and focuses solely on style aspects by using the style attention head as $\Psi(\cdot)$.

In content-style composition (content described by both text and a reference content image; style described by a reference style image), the AFA module injects content (extracted from the reference content) and style features (from the reference style image) separately using their respective attention heads in the ViT (Somepalli et al., 2024). The SOC module controls *content* by minimizing the discrepancy between content features from the generated image and the *reference content* image, and *style* by minimizing the discrepancy between style features extracted from the generated and *reference style* image. This distinction from prior works enables our method to prevent leakage.

## 5   THEORETICAL JUSTIFICATIONS

**Problem setup:** We outline an approach to derive the optimal controller for a special case of our control problem (2). We substitute $t \leftarrow 1 - t$ to account for the time reversal in the reverse-SDE (1). Here, $X_0^u \sim \mathcal{N}(0, \mathrm{I}_d)$ and $X_1^u \sim p_{data}$. We consider the dynamic without the Brownian motion: $\mathrm{d}X_t^u = v(X_t^u, u, t)\mathrm{d}t$, $X_{t_0}^u = \mathbf{x}_0$, where $0 \leq t_0 \leq t \leq t_N \leq 1$ and $v : \mathbb{R}^d \times \mathbb{R}^d \times [t_0, t_N] \to \mathbb{R}^d$ denotes the drift field. The optimal controller $u^*$ can be derived by solving the Hamilton-Jacobi-Bellman (HJB) equation (Fleming & Rishel, 2012; Basar et al., 2020), see Appendix A for details.

**Incorporating optimal control in diffusion:** Following recent works (Kappen, 2008; Chen et al., 2023), we consider a dynamical system whose drift field minimizes a transient trajectory cost and a terminal cost (weighted by $\gamma$) to ensure "closeness" to reference content $x_1$ (Appendix A.1). **Proposition A.2** (Chen et al., 2023) outlines the optimal control in the limiting setting where $\gamma \to \infty$. Furthermore, suppose we replace $x_1$ with its conditional expectation (discussed in Remark A.3), *the resulting dynamic is the standard reverse-SDE for the Orstein-Uhlenbeck (OU) diffusion process for a particular noise schedule.* This connection between classic linear quadratic control and the standard reverse-SDE allows us to study other diffusion problems (*e.g.*, personalization) through the lens of stochastic optimal control. For instance, we derive the optimal controller given reference *style features* $y_1$ at the terminal time.

**Proposition 5.1.** *Suppose $A \in \mathbb{R}^{k \times d}$ be a linear style extractor that operates on the terminal state $X_1^u \in \mathbb{R}^d$. Given reference style features $y_1$, consider the control problem:*

$$\min_{u \in \mathcal{U}} \int_{t_0}^1 \frac{1}{2} \|u(X_t^u, t)\|^2 \, dt + \frac{\gamma}{2} \|AX_1^u - y_1\|_2^2, \text{ where } \mathrm{d}X_t^u = u(X_t^u, t) \, \mathrm{d}t, \, X_{t_0}^u = x_0.$$

*Then, in the limit when $\gamma \to \infty$, the optimal controller $u^* = \frac{(A^T A)^{-1} A^T (y_1 - A\mathbf{x}_t)}{1 - t}$, which yields the following controlled dynamic:* $\mathrm{d}X_t^u = \frac{(A^T A)^{-1} A^T (y_1 - A\mathbf{x}_t)}{1 - t} \mathrm{d}t.$

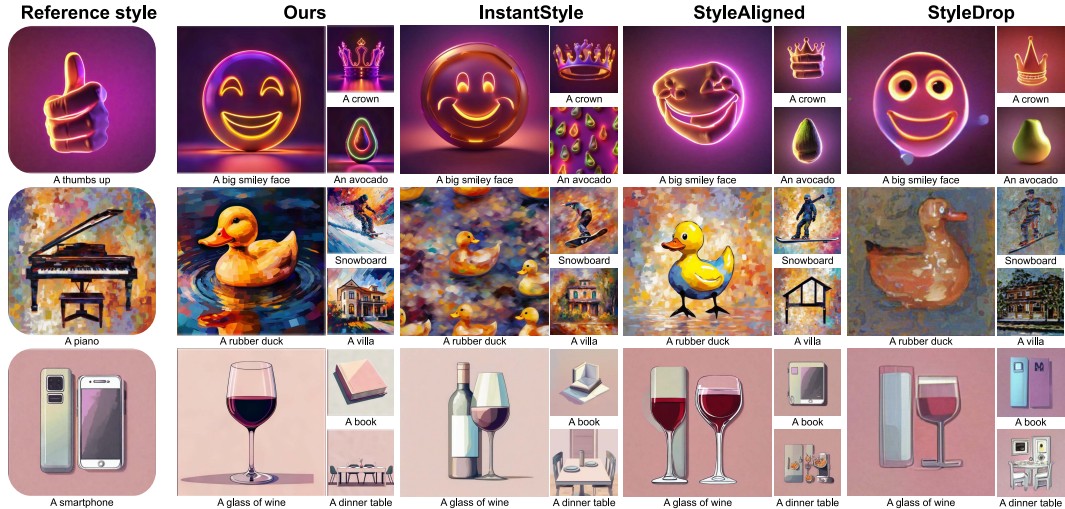

Figure 2: **Qualitative results for stylization:** A comparison with state-of-the-art methods (InstantStyle (Wang et al., 2024a), StyleAligned (Hertz et al., 2023), StyleDrop (Sohn et al., 2023)) highlights our advantages in preventing information leakage from the reference style and adhering more closely to desired prompts.

**Implication.** The optimal controller depends on the reference *style features* $y_1$ at the terminal time, instead of the image content encoded in $x_1$. To simulate the controlled dynamic in practice, we use CSD (Somepalli et al., 2024) as a style feature extractor and replace $y_1$ with the style features extracted from the expected terminal state $\mathbb{E}[X_1^u|X_t^u]$, as discussed in **Appendix A.2**.

**Drift modulation through optimal controller:** We then study a control problem where the velocity field is a linear combination of the state and the control variable. This problem is interesting to study because the reverse-SDE dynamic of the standard OU process has a drift field of the form: $v(X_t,t) = -X_t - 2\nabla \log p(X_t,t)$. For a Gaussian prior $X_0 \sim \mathcal{N}(0,\mathrm{I})$, the law of the OU process satisfies $\nabla \log p(X_t,t) = -X_t$, and the corresponding drift field becomes $v(X_t,t) = X_t$. Our goal is to modulate this drift field using a controller $u(X_t^u,t)$. The result below provides the structure of the optimal control (again in the setting where the terminal objective is known; see Appendix A1).

**Proposition 5.2.** *Suppose $A \in \mathbb{R}^{k \times d}$ be a linear style extractor that operates on the terminal state $X_1^u \in \mathbb{R}^d$. Let $\mathbf{p}_t$ denote $\nabla_{\mathbf{x}} V^*(\mathbf{x},t)$ in HJB equation (A.1). Given reference style features $y_1$, consider the control problem:*

$$\min_{u \in \mathcal{U}} \int_{t_0}^{1} \frac{1}{2} \|u(X_t^u,t)\|^2 \, dt + \frac{\gamma}{2} \|AX_1^u - y_1\|_2^2, \text{ where } \mathrm{d}X_t^u = [X_t^u + u(X_t^u,t)] \, \mathrm{d}t, \ X_{t_0}^u = x_0,$$

*Then, the optimal controller becomes $u^*(t) = -\mathbf{p}_t$, where the instantaneous state $X_t^u = \mathbf{x}_t$ and $\mathbf{p}_t$ satisfy the following coupled transitions:*

$$\begin{bmatrix} \mathbf{x}_t \\ \mathbf{p}_t \end{bmatrix} = \begin{bmatrix} x_0 e^t - \frac{\gamma}{2} A^T (A\mathbf{x}_1 - y_1) e^{1+t} + \frac{\gamma}{2} A^T (A\mathbf{x}_1 - y_1) e^{1-t} \\ \gamma A^T (A\mathbf{x}_1 - y_1) e^{1-t} \end{bmatrix}.$$

**Summary.** We build on the connection between optimal control and reverse diffusion (see Appendices A.1-A.3 for details). The general strategy is to derive the optimal controller with known terminal state, and then replace the terminal state in the controller with its estimate using Tweedie's formula. For stylized models and Gaussian prior, the controllers have an explicit form. However in practice, the data distribution may not be Gaussian, and thus, we do not aim for a closed-form expression to modulate the drift. This line of analysis, however, points to our method RB-Modulation. As discussed in §4, we incorporate a style descriptor in our controller's terminal cost and evaluate the resulting drift at each reverse time step either through back propagating through the score network (**Algorithm 1**), or an approximation based on proximal gradient updates (**Algorithm 2**).

## 6 EXPERIMENTS

**Metrics:** Evaluating stylized synthesis is challenging due to the subjective nature of style, making simple metrics inadequate. We follow a two step approach: first using metrics from prior

Table 1: **User study:** We report the % of human preference on ours *vs.* alternatives for overall quality (OQ), style alignment (SA), and prompt alignment (PA), including ties where users couldn't decide. Our method consistently outperforms alternatives, achieving higher scores in all metrics.

| Human | **Ours *vs*. InstantStyle** | | | **Ours *vs*. StyleAligned** | | | **Ours *vs*. IP-Adapter** | | |
|---|---|---|---|---|---|---|---|---|---|
| Preference (%) | OQ ↑ | SA ↑ | PA ↑ | OQ ↑ | SA ↑ | PA ↑ | OQ ↑ | SA ↑ | PA ↑ |
| **Alternative** | 39.8 | 38.5 | 39.5 | 24.4 | 27.8 | 29.4 | 8.1 | 20.1 | 8.3 |
| **Tie** | 9.3 | 6.4 | 7.3 | 8.8 | 7.1 | 5.8 | 6.9 | 4.8 | 4.5 |
| **RB-Modulation** (ours) | **51.0** | **55.1** | **53.3** | **66.9** | **65.1** | **64.9** | **85.0** | **75.1** | **87.2** |

works and then conducting human evaluation. To evaluate prompt-image alignment, we use CLIP-T score (Hertz et al., 2023; Sohn et al., 2023; Wang et al., 2024a) and ImageReward (Xu et al., 2024), which also consider human aesthetics, distortions, and object completeness. When a style description is provided, CLIP-T and ImageReward also capture style alignment. We assess style similarity using DINO (Caron et al., 2021) and content similarity using CLIP-I (Radford et al., 2021) as in prior work (Hertz et al., 2023; Ruiz et al., 2023; Sohn et al., 2023), and highlight their limitations in disentangling style and content performance in evaluation. Given the importance of human evaluation in T2I personalization (Hertz et al., 2023; Sohn et al., 2023; Ruiz et al., 2023; Shah et al., 2023; Jeong et al., 2024), we also conduct a user study though Amazon Mechanical Turk to measure both style and text alignment.

**Datasets and baselines:** We use style images from StyleAligned benchmark (Hertz et al., 2023) for stylization and content images from DreamBooth (Ruiz et al., 2023) for content-style composition. We base RB-Modulation on the recently released StableCascade (Pernias et al., 2024). We compare with three training-free methods: InstantStyle (Wang et al., 2024a) (state-of-the-art), IP-Adapter (Ye et al., 2023), and StyleAligned (Hertz et al., 2023). For completeness, we also compare with training-based methods StyleDrop (Sohn et al., 2023) and ZipLoRA (Shah et al., 2023).

**Implementation details:** All experiments run on a single A100 NVIDIA GPU. We use the same hyper-parameters for our method across tasks, and default settings for alternative methods as per their original papers. Details are provided in Appendix B.1.

## 6.1 IMAGE STYLIZATION

**Qualitative analysis:** This section describes image stylization experiments using a text prompt and a reference style image. Figure 2 compares our method with SoTA **training-free** InstantStyle (Wang et al., 2024a) and StyleAligned (Hertz et al., 2023), and **training-based** StyleDrop (Sohn et al., 2023). Except for StyleDrop, which requires ∼5 minutes of training per style, all methods, including ours, are training-free and complete inference in <1 minute. While all methods produce reasonable outputs, alternative methods encounter issues with information leakage. For instance, in the third row of Figure 2, StyleAligned and StyleDrop generate a wine bottle and book resembling the smartphone in the reference style image. In the last row, StyleAligned leaks the house and the background of the reference image; InstantStyle exhibits color leakage from the house, resulting in similar-colored images. Our method accurately adheres to the prompt in the desired style. As illustrated in the second and the third row, our method generates only one glass of wine and a high-fidelity rubber duck, compared to baselines where extra items appear (wine bottles styled like the left smartphone) or incorrect styles (cartoon-style rubber duck).

**User study:** Given the subjective nature of this field, we conduct a user study on Amazon Mechanical Turk with 155 participants using 100 styles from the StyleAligned dataset (Hertz et al., 2023), collecting a total of 7,200 answers (8 responses for each question). Each user answers 3 questions comparing our method with an alternative method regarding (1) overall quality, (2) style alignment, and (3) prompt alignment (details in the Appendix B.8). Table 1 summarizes the percentage of human preferences for our method, the alternative method, or a tie. Our method consistently outperforms the alternatives, including the current SoTA method InstantStyle (Wang et al., 2024a). The preference rates over all three metrics highlight the effectiveness of our method RB-Modulation.

**Quantitative analysis:** Table 2 evaluates 300 prompts and 100 styles on the StyleAligned dataset (Hertz et al., 2023) using three metrics, with and without style descriptions in the prompts. Our method outperforms others notably in the ImageReward metric, closely matching human aesthetics assessment from the user study in Table 1. In addition, the CLIP-T score indicates our effective alignment between generated images and text prompts. While IP-Adapter and StyleAligned

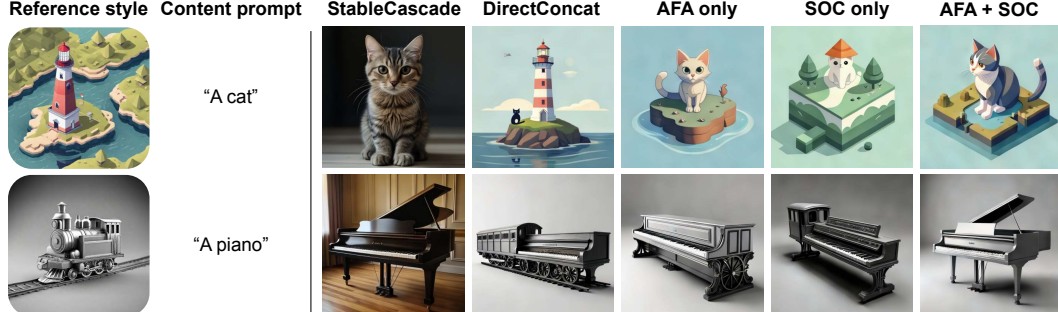

Figure 3: **Ablation study:** We show the effectiveness of our different proposed components by sequentially adding them to vanila StableCascade (Pernias et al., 2024). DirectConcat involves concatenating reference image embeddings with prompt embeddings.

Table 2: **Quantitative results for stylization:** We compare alternative methods on three metrics: ImageReward (Xu et al., 2024) and CLIP-T (Radford et al., 2021) for prompt alignment, DINO (Caron et al., 2021) for style alignment. Note that DINO score does not capture information leakage, so higher scores are not necessarily better (§B.5).

| With style description? | ImageReward ↑ | | CLIP-T score ↑ | | DINO score | |
|---|---|---|---|---|---|---|
| | No | Yes | No | Yes | No | Yes |
| **IP-Adapter** (Ye et al., 2023) | -1.99 | -1.51 | 0.21 | 0.26 | 0.89 | 0.89 |
| **StyleAligned** (Hertz et al., 2023) | -0.68 | 0.01 | 0.26 | 0.31 | 0.80 | 0.85 |
| **InstantStyle** (Wang et al., 2024a) | 0.09 | 0.72 | 0.29 | 0.33 | 0.68 | 0.72 |
| **RB-Modulation** (ours) | 0.91 | 1.18 | 0.30 | 0.34 | 0.68 | 0.73 |

have higher DINO scores, their lower rating in ImageReward, CLIP-T and user preference expose information leakage from the reference style images. Nevertheless, our DINO score remains competitive with the leading method InstantStyle. Notably, all metrics show improvement with style descriptions, particularly in ImageReward, where leveraging style descriptions enhances prompt alignment. Our method achieves high ImageReward and CLIP-T score even without style descriptions, suggesting robustness in prompt alignment without explicit style information in the prompt.

**Ablation Study:** Figure 3 shows an ablation study of the AFA and SOC modules. We include a baseline, "DirectConcat", which concatenates reference style embeddings with text embeddings in the cross-attention modules. DirectConcat mixes both embeddings, making it less effective in disentangling style from prompts (*e.g.*, cat *vs.* lighthouse). While AFA or SOC alone mitigates this by modulating the reverse drift and attention modules (§4), each has drawbacks. AFA alone fails to capture the cat's style accurately, and SOC alone misplaces elements, like "a lighthouse hat on the cat" and "a railroad trunk on a piano". We observe consistent improvements with each module, with the best results when combined.

## 6.2 Content-Style Composition

Since this paper primarily focuses on style-based personalization, we perform extensive experiments on stylization. To further demonstrate the versatility of our framework, we also explore content-style composition as an additional capability.

**Qualitative analysis:** Content-style composition aims to preserve the essence of both content and style depicted in the reference images, while ensuring the resulting image aligns with a given text prompt. Figure 4 compares our method against **training-free** InstantStyle (Wang et al., 2024a), IP-Adapter (Ye et al., 2023), and **training-based** ZipLoRA (Shah et al., 2023). Notably, the training-free InstantStyle and IP-Adapter rely on ControlNet (Zhang et al., 2023), which often constrains their ability to accurately follow prompts for changing the pose of the generated content, such as illustrating "dancing" in Figure 4(b), or "walking" in (c). In contrast, our method avoids the need for ControlNet or adapters, and can effectively capture the distinctive attributes of both style and content images while adhering to the prompt to generate diverse images. In Figure 4(a), our method accurately captures elements like "table" and "river" that are overlooked in InstantStyle and IP-Adapter. In addition, our method mitigates information leakage, as evidenced in Figure 4(b), where the trunk of the tree behind the sloth is erroneously captured by InstantStyle and IP-Adapter but not

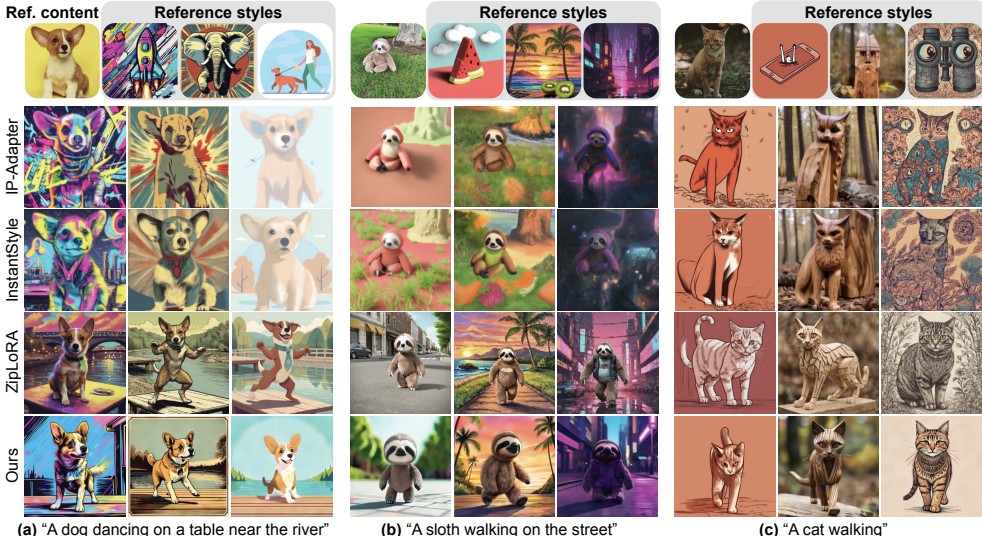

Figure 4: **Qualitative results for content-style composition:** Our method shows better prompt alignment and greater diversity than training-free methods IP-Adapter (Ye et al., 2023) and InstantStyle (Wang et al., 2024a), and have competitive performance with training-based ZipLoRA (Shah et al., 2023).

Table 3: **Quantitative results for composition:** In addition to stylization metrics, we use CLIP-T score (Radford et al., 2021) to evaluate content alignment with the reference image. Similar to DINO, CLIP-I could inflate test score (Sohn et al., 2023; Shah et al., 2023) due to content leakage, but does not correlate to user preference; higher scores do not indicate better human preference.

|  | ImageReward ↑ | CLIP-T score ↑ | DINO score | CLIP-I score |
|---|---|---|---|---|
| **IP-Adapter** | -0.78 | 0.22 | 0.73 | 0.68 |
| **InstantStyle** | -0.54 | 0.21 | 0.71 | 0.71 |
| **RB-Modulation** (ours) | 0.74 | 0.26 | 0.74 | 0.71 |

by ours. Compared to ZipLoRA (Shah et al., 2023) that requires training of 12 LoRAs (Hu et al., 2021) and additional merge layers for each composition, our method requires no training at all while yielding competitive or better results. For instance, our method effectively captures the 2D cartoon and 3D rendering styles as illustrated in Figures 4(a) and (b).

**Quantitative analysis:** Table 3 shows quantitative evaluation using 50 styles from StyleAligned dataset (Hertz et al., 2023) and 5 contents from DreamBooth dataset (Ruiz et al., 2023). Unlike prior works (Hertz et al., 2023; Sohn et al., 2023; Shah et al., 2023; Ruiz et al., 2023; Jeong et al., 2024) reporting either DINO and CLIP-I scores, we present both metrics and demonstrate comparable performance across them. Additionally, we obtain notably higher ImageReward score, which aligns closely with human aesthetics assessment as evidenced in §6.1 and (Xu et al., 2024). Consequently, we omitted a user study in this section. For more details, please refer to Appendix B.1.

## 7 CONCLUSION

We introduced Reference-Based modulation (RB-Modulation), a test-time optimization method for personalizing transformer-based diffusion models. RB-Modulation builds on concepts from stochastic optimal control to modulate the drift field of reverse diffusion dynamics, incorporating desired attributes (*e.g.*, style or content) via a terminal cost. Our Attention Feature Aggregation (AFA) module decouples content and style in the cross-attention layers and enables precise control over both. In addition, we derived theoretical connections between linear quadratic control and the denoising diffusion process, which led to the creation of RB-Modulation. Empirically, our method outperformed current state-of-the-art methods in stylization and content-style composition. To our best knowledge, this is the first training-free personalization framework using stochastic optimal control, which marks the departure from external adapters or ControlNets.

## 8    BROADER IMPACT STATEMENT

**Social impact:** Image stylization and content-style composition based on diffusion models potentially have both positive and negative social impact. This technology provides an easy-to-use tool to the general public for image generation which can help visualize their artistic ideas. On the other hand, our work on stylization and content-style composition poses a risk of generating arts that closely mimic or infringe upon existing copyrighted material, leading to legal and ethical issues. More broadly, our method inherits the risks from T2I models which are capable of generating fake contents that can be misused by malicious users.

**Safeguards:** We build on StableCascade (Pernias et al., 2024), which has a mechanism to filter offensive image generations. Our framework RB-Modulation inherits these safeguards. In addition, to mitigate misuse, we believe it is crucial to ensure the underlying model's safety, which may involve (i) watermarking AI-generated artworks and (ii) implementing an NSFW filter to remove inappropriate contents.

**Reproducibility:** The pseudocode and hyper-parameter details have been provided in the paper. The source code is available on the project page: https://rb-modulation.github.io/.

## ACKNOWLEDGMENTS

This research has been supported by NSF Grant 2019844, a Google research collaboration award, and the UT Austin Machine Learning Lab. Litu Rout has been supported by Ju-Nam and Pearl Chew Presidential Fellowship and George J. Heuer Graduate Fellowship from UT Austin.

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

# A    ADDITIONAL THEORETICAL RESULTS

In this section, we restate the propositions more precisely and provide their technical proofs. First, we recall standard terminologies from optimal control literature (Fleming & Rishel, 2012). For $0 \leq t_0 \leq t \leq t_N \leq 1$, the cost function associated with the controller $u(\cdot)$ is defined by the integral:

$$V(u; \mathbf{x}_0, t_0) = \int_{t_0}^{t_N} \ell\left(X_t^u, u, t\right) dt + h\left(X_{t_N}^u\right), \quad X_{t_0}^u = \mathbf{x}_0, \tag{4}$$

where $\ell(\cdots)$ denotes a scalar valued function of the state $X_t^u$, controller $u(\cdot)$, and instantaneous time $t$. The value function $V^*(\mathbf{x}_0, t_0)$ is defined as the minimum value of $V(u; \mathbf{x}_0, t_0)$ over the set of admissible controllers $\mathcal{U}$, i.e.,

$$V^* = V^*(\mathbf{x}_0, t_0) = \min_{u \in \mathcal{U}} V(u; \mathbf{x}_0, t_0) = \min_{u \in \mathcal{U}} \int_{t_0}^{t_N} \ell\left(X_t^u, u, t\right) dt + h\left(X_{t_N}^u\right), \quad X_{t_0}^u = \mathbf{x}_0, \tag{5}$$

which satisfies a Partial Differential Equation (PDE) given below in **Theorem A.1**.

**Theorem A.1** (HJB Equation, (Fleming & Rishel, 2012; Basar et al., 2020)). *If $V^*$ has continuous partial derivatives, then it must satisfy the following PDE, also known as Hamilton-Jacobi-Bellman (HJB) equation:*

$$-\frac{\partial V^*}{\partial t}(\mathbf{x}, t) = \min_{u \in \mathcal{U}} \left[ H\left(\mathbf{x}, \nabla_{\mathbf{x}} V^*(\mathbf{x}, t), u, t\right) := \ell(\mathbf{x}, u, t) + \left(\nabla_{\mathbf{x}} V^*(\mathbf{x}, t)\right)^T v(\mathbf{x}, u, t) \right].$$

*Also, the Hamiltonian $H\left(\mathbf{x}, \nabla_{\mathbf{x}} V^*(\mathbf{x}, t), u, t\right)$, optimal controller $u^*(t)$ and the state trajectory $\mathbf{x}^*(t)$ must satisfy*

$$\min_{u \in \mathcal{U}} H\left(\mathbf{x}^*(t), \nabla_{\mathbf{x}} V^*(\mathbf{x}^*(t), t), u, t\right) = H\left(\mathbf{x}^*(t), \nabla_{\mathbf{x}} V^*(\mathbf{x}^*(t), t), u^*(t), t\right).$$

## A.1    INTERPRETING REVERSE-SDE AS A SOLUTION TO OPTIMAL CONTROL

For clarity, we restate the problem setup here and describe the main ideas from §4 in more details. **Problem setup:** We discuss a standard approach to derive the optimal controller in a special case of our control problem (2). We substitute $t \leftarrow 1 - t$ to account for the time reversal in the reverse-SDE (1). In this setup, $X_0^u \sim \mathcal{N}(0, \mathrm{I}_d)$ and $X_1^u \sim p_{data}$. We consider the following dynamic without the Brownian motion:

$$\mathrm{d}X_t^u = v(X_t^u, u, t)\mathrm{d}t, \quad X_{t_0}^u = \mathbf{x}_0, \tag{6}$$

where $0 \leq t_0 \leq t \leq t_N \leq 1$ and $v : \mathbb{R}^d \times \mathbb{R}^d \times [t_0, t_N] \to \mathbb{R}^d$ denotes the drift field. The optimal controller $u^*$ can be derived by solving the Hamilton-Jacobi-Bellman (HJB) equation (Fleming & Rishel, 2012; Basar et al., 2020), see Appendix A for details. By certainty equivalence (when the drift and diffusion coefficients are linear time-varying (Astrom, 1971), which occurs when $p_{data}$ is Gaussian; see also discussion in Section A.3), the same $u^*$ applies to a more general case with the Brownian motion (Chen et al., 2023), where

$$\mathrm{d}X_t^u = v(X_t^u, u, t)\mathrm{d}t + \mathrm{d}W_t, \quad X_{t_0}^u = \mathbf{x}_0. \tag{7}$$

Therefore, we analyze the reverse dynamic in the absence of the Brownian motion, and employ the same controller in more general cases with the Brownian motion.

Below, we consider a dynamical system whose drift field is chosen to minimize a transient trajectory cost and a terminal cost (weighted by $\gamma$) that enforces "closeness" to reference content $x_1$. **Proposition A.2** provides the structure of the optimal control in the limiting setting where $\gamma \to \infty$. Furthermore, suppose we replace $x_1$ with its conditional expectation (discussed in Remark A.3), the resulting dynamic, interestingly, is the standard reverse-SDE for the Orstein-Uhlenbeck (OU) diffusion process. This connection between optimal control (more precisely, classic Linear Quadratic Control) and the standard reverse-SDE provides us a path to study other diffusion problems (*e.g.* personalization (Ruiz et al., 2023; Hertz et al., 2023; Sohn et al., 2023; Wang et al., 2024a), image editing or inversion (Mokady et al., 2023; Delbracio & Milanfar, 2023; Rout et al., 2023b; 2024; 2023a)) through the lens of stochastic optimal control.

**Proposition A.2** (Linear optimal control with quadratic cost (Chen et al., 2023)). *Consider the control problem:*

$$\min_{u \in \mathcal{U}} \int_{t_0}^1 \frac{1}{2} \|u(X_t^u, t)\|^2 \, dt + \frac{\gamma}{2} \|X_1^u - x_1\|_2^2,$$
$$\text{where } \mathrm{d}X_t^u = u(X_t^u, t) \, \mathrm{d}t, \quad X_{t_0}^u = x_0$$

*Then, in the limit when $\gamma \to \infty$, the optimal controller is given by $u^* = \frac{x_1 - X_t^u}{1-t}$, which yields $\mathrm{d}X_t^u = \frac{x_1 - X_t^u}{1-t}\mathrm{d}t$ for the deterministic case and $\mathrm{d}X_t^u = \frac{x_1 - X_t^u}{1-t}\mathrm{d}t + \mathrm{d}W_t$ for the stochastic case.*

The optimal controller for the problem presented in **Proposition A.2** can be derived using established techniques from control theory (Fleming & Rishel, 2012; Basar et al., 2020; Kappen, 2008); the specific form of the above result follows from (Chen et al., 2023) (but without their momentum term). The key steps in this derivation include: (1) computing the Hamiltonian, (2) applying the minimum principle theorem to derive a set of differential equations, and (3) taking the limit as $\gamma \to \infty$. These three steps are fundamental in deriving a closed-form solution. The final step is critical for satisfying hard terminal constraint and is essential for the practical implementation of **Algorithm 1** and **Algorithm 2**, as detailed in §4.

For generative modeling, the controlled dynamics described in **Proposition A.2** cannot be directly applied. This limitation arises because the optimal control $u^*$ depends on the terminal state $x_1$, making it non-causal or reliant on future information. Inspired by recent advancements in flow-based generative models (Lipman et al., 2022; Liu et al., 2022), we make the optimal controller causal by replacing the terminal state with its conditional expectation given the current state, i.e., , *i.e.* $x_1 \leftarrow \mathbb{E}[X_1^u | X_t^u = \mathbf{x}_t]$. This modification results in a controlled dynamic that can be simulated to produce a generative model incorporating principles from optimal control, as elaborated in **Remark A.3**.

**Remark A.3** (Connections between diffusion-based generative modeling and stochastic optimal control). *Following conditional diffusion models and optimal transport paths (Lipman et al., 2022; Liu et al., 2022), where $X_t^f = tX_0^f + (1-t)\epsilon$, the state variable $X_t^u$ is equal in distribution to $X_{1-t}^f = (1-t)X_0^f + t\epsilon$, $\epsilon \sim \mathcal{N}(0, \mathrm{I}_d)$ after time reversal. Now, we use Tweedie's formula (Efron, 2011) to compute the posterior mean:*

$$\mathbb{E}[X_1^u | X_t^u] = \frac{X_t^u}{1-t} + \frac{t^2}{1-t}\nabla \log p(X_t^u, 1-t). \tag{8}$$

*Substituting the posterior mean in the controlled reverse dynamic of **Proposition A.2**, we arrive at*

$$\mathrm{d}X_t^u = \frac{(\mathbb{E}[X_1^u | X_t^u] - X_t^u)}{(1-t)}\mathrm{d}t + \mathrm{d}W_t$$
$$= \left[\frac{t}{(1-t)^2}X_t^u + \frac{t^2}{(1-t)^2}\nabla \log p(X_t^u, 1-t)\right]\mathrm{d}t + \mathrm{d}W_t.$$

We observe that the above equation is structurally the same as reverse-SDE associated with a forward Orstein-Uhlenbeck (OU) diffusion process. This relation between diffusion-based generative models and optimal control is further explored in the Appendices below.

Indeed, diffusion models (Ho et al., 2020; Song et al., 2021b; Rombach et al., 2022; Podell et al., 2023; Pernias et al., 2024) provide an effective approximation to the terminal state of a denoising process. This approximation has been used for a variety of generative modeling tasks. Also, the terminal state can be approximated using Tweedie's formula (Efron, 2011) with a learned score function (Ho et al., 2020)[1]. By utilizing these pre-trained diffusion models, we can employ the connection to optimal control as discussed above to develop practically implementable generative models that incorporates terminal objectives such as style and personalization. Consequently, the subsequent sections are dedicated to deriving the optimal controller assuming a known terminal state; we will approximate this in practice using Tweedie's formula as above.

---

[1]Alternatively, when the reverse process is described by a probability flow ODE, a trained neural network can directly predict the terminal state (Song et al., 2021a).

## A.2 INCORPORATING PERSONALIZED STYLE CONSTRAINTS THROUGH A TERMINAL COST

In this section, we derive the optimal controller when we have access to the reference *style features* $y_1$ at the terminal time (instead of the content of the image encoded through $x_1$).

**Proposition A.4.** *Suppose $A \in \mathbb{R}^{k \times d}$ be a linear style extractor that operates on the terminal state $X_1^u \in \mathbb{R}^d$. Given reference style features $y_1$, consider the control problem:*

$$\min_{u \in \mathcal{U}} \int_{t_0}^1 \frac{1}{2} \|u(X_t^u, t)\|^2 \, dt + \frac{\gamma}{2} \|AX_1^u - y_1\|_2^2, \tag{9}$$

$$\text{where } \mathrm{d}X_t^u = u(X_t^u, t) \, \mathrm{d}t, \quad X_{t_0}^u = x_0, \tag{10}$$

*Then, in the limit when $\gamma \to \infty$, the optimal controller $u^* = \frac{\left(A^T A\right)^{-1} A^T (y_1 - AX_t^u)}{1-t}$, which yields the following controlled dynamic:*

$$\mathrm{d}X_t^u = \frac{\left(A^T A\right)^{-1} A^T \left(y_1 - AX_t^u\right)}{1 - t} \mathrm{d}t. \tag{11}$$

*Proof.* We derive the closed-form solution of the optimal controller given a fixed terminal state condition. This is similar to (Chen et al., 2023), where the reverse process is accelerated using momentum (see also (Kappen, 2008; Basar et al., 2020) for further details on this approach). The distinction, however, lies in the treatment of the terminal constraint. For completeness, we provide full details of the proof below.

To derive the closed-form solution[2], recall from equation (5) that $\ell(\mathbf{x}_t, \mathbf{u}_t, t) = \frac{1}{2} \|\mathbf{u}_t\|^2$ and the terminal cost $h(\mathbf{x}_1) = \frac{\gamma}{2} \|A\mathbf{x}_1 - y_1\|^2$. Let $\mathbf{p}_t$ represent $\nabla_{\mathbf{x}} V^*(\mathbf{x}, t)$ in **Theorem A.1**. Then, the Hamiltonian of the control problem (9) is given by

$$H(\mathbf{x}_t, \mathbf{p}_t, \mathbf{u}_t, t) = \ell(\mathbf{x}_t, \mathbf{u}_t, t) + \mathbf{p}_t^T \mathbf{u}_t$$

$$= \frac{1}{2} \|\mathbf{u}_t\|^2 + \mathbf{p}_t^T \mathbf{u}_t.$$

Since the minimizer of the Hamiltonian is $\mathbf{u}_t^* = -\mathbf{p}_t$, the value function becomes

$$V^* = \min_{\mathbf{u}_t} H(\mathbf{u}_t, \mathbf{p}_t, \mathbf{u}_t, t) = H(\mathbf{u}_t, \mathbf{p}_t, \mathbf{u}_t^*, t) = -\frac{1}{2} \|\mathbf{p}_t\|^2. \tag{12}$$

Now, we use minimum principle theorem (Basar et al., 2020) to obtain the following set of differential equations:

$$\frac{\mathrm{d}\mathbf{x}_t}{\mathrm{d}t} = \nabla_{\mathbf{p}} H\left(\mathbf{x}_t, \mathbf{p}_t, \mathbf{u}_t^*, t\right) = -\mathbf{p}_t; \tag{13}$$

$$\frac{\mathrm{d}\mathbf{p}_t}{\mathrm{d}t} = -\nabla_{\mathbf{x}} H\left(\mathbf{x}_t, \mathbf{p}_t, \mathbf{u}_t^*, t\right) = 0; \tag{14}$$

$$\mathbf{x}_{t_0} = x_0; \tag{15}$$

$$\mathbf{p}_{t_N} = \nabla_{\mathbf{x}} h\left(\mathbf{x}_{t_N}, t_N\right) = \gamma A^T \left(A\mathbf{x}_{t_N} - y_1\right). \tag{16}$$

Integrating both sides of (13), we have

$$\int_{t_0}^1 \mathrm{d}\mathbf{x}_t = -\int_{t_0}^1 \mathbf{p}_t \mathrm{d}t = -\mathbf{p}\left(1 - t_0\right), \tag{17}$$

where the last equality is due to (14), which states that $\mathbf{p}_t$ is a constant independent of time $t$. This implies $\mathbf{x}_1 = \mathbf{x}_{t_0} - \mathbf{p}(1 - t_0)$. From (16), we know for $t_N = 1$ that

$$\mathbf{p}_1 = \gamma A^T \left(A\mathbf{x}_1 - y_1\right)$$

$$= \gamma \left(A^T A \left(x_0 - \mathbf{p}(1 - t_0)\right) - A^T y_1\right)$$

$$= \gamma A^T A x_0 - \gamma A^T A \mathbf{p}_1 (1 - t_0) - \gamma A^T y_1 \tag{18}$$

---

[2] With slight abuse of notation, we use $\mathbf{x}_t$ to denote $X_t^u$ and $\mathbf{u}_t$ to denote $u(X_t^u, t)$ in the deterministic case.

Rearranging (18) and solving for $\mathbf{p}_1$, we get

$$\mathbf{p}_1 = \gamma \left( I + \gamma A^T A \left( 1 - t_0 \right) \right)^{-1} \left( A^T A x_0 - A^T y_1 \right)$$

$$= \left( \frac{I}{\gamma} + A^T A \left( 1 - t_0 \right) \right)^{-1} \left( A^T A x_0 - A^T y_1 \right) = \mathbf{p} \tag{19}$$

Passing (19) through the limit $\gamma \to \infty$, we get

$$\lim_{\gamma \to \infty} \mathbf{p} = \frac{\left( A^T A \right)^{-1} \left( A^T A x_0 - A^T y_1 \right)}{1 - t_0}. \tag{20}$$

Therefore, the optimal control becomes $\mathbf{u}_t^* = -\mathbf{p} = -\frac{\left( A^T A \right)^{-1} \left( A^T A \mathbf{x}_t - A^T y_1 \right)}{1 - t}$, and the resulting dynamical system is given by

$$\mathrm{d}\mathbf{x}_t = \frac{\left( A^T A \right)^{-1} A^T \left( y_1 - A\mathbf{x}_t \right)}{1 - t} \mathrm{d}t,$$

for the deterministic process and

$$\mathrm{d}\mathbf{x}_t = \frac{\left( A^T A \right)^{-1} A^T \left( y_1 - A\mathbf{x}_t \right)}{1 - t} \mathrm{d}t + \mathrm{d}W_t,$$

for the stochastic process with the Brownian motion. This completes the statement of the proof. $\square$

**Implications:** The optimal controller depends on the reference *style features* $y_1$ at the terminal time (instead of the image content $x_1$ as in Appendix A.1). The reverse dynamic can be simulated in practice by using CSD (Somepalli et al., 2024) as a style feature extractor and replacing $y_1$ with the extracted style features from the expected terminal state $\mathbb{E}[X_1^u | X_t^u]$, as discussed in **Remark A.3**. This makes the controller drift causal and non-anticipating future information

## A.3 INCORPORATING STYLE THROUGH MODULATION AND A TERMINAL COST

In this section, we study a control problem where the velocity field is a linear combination of the state and the control variable. This problem is interesting to study because of the following reason. The reverse-SDE dynamic of the standard OU process has a drift field of the form:

$$v\left( X_t, t \right) = -X_t - 2\nabla \log p(X_t, t).$$

For a Gaussian prior $X_0 \sim \mathcal{N}\left( 0, \mathrm{I} \right)$, the law of the OU process satisfies $\nabla \log p\left( X_t, t \right) = -X_t$, and the corresponding drift field becomes $v\left( X_t, t \right) = X_t$. Our goal is to modulate this drift field using a controller $u\left( X_t^u, t \right)$. The result below provides the structure of the optimal control (again in the setting where the terminal objective is known; see Appendix A1).

**Proposition A.5.** *Suppose $A \in \mathbb{R}^{k \times d}$ be a linear style extractor that operates on the terminal state $X_1^u \in \mathbb{R}^d$. Let $\mathbf{p}_t$ denote $\nabla_{\mathbf{x}} V^*(\mathbf{x}, t)$ in HJB equation (A.1). Given reference style features $y_1$, consider the control problem:*

$$\min_{u \in \mathcal{U}} \int_{t_0}^{1} \frac{1}{2} \|u(X_t^u, t)\|^2 \, dt + \frac{\gamma}{2} \|AX_1^u - y_1\|_2^2, \tag{21}$$

$$\text{where } \mathrm{d}X_t^u = \left[ X_t^u + u(X_t^u, t) \right] \mathrm{d}t, \quad X_{t_0}^u = x_0, \tag{22}$$

*Then, the optimal controller becomes $u^*(t) = -\mathbf{p}_t$, where the instantaneous state $X_t^u = \mathbf{x}_t$ and $\mathbf{p}_t$ satisfy the following:*

$$\begin{bmatrix} \mathbf{x}_t \\ \mathbf{p}_t \end{bmatrix} = \begin{bmatrix} x_0 e^t - \frac{\gamma}{2} A^T \left( A\mathbf{x}_1 - y_1 \right) e^{1+t} + \frac{\gamma}{2} A^T \left( A\mathbf{x}_1 - y_1 \right) e^{1-t} \\ \gamma A^T \left( A\mathbf{x}_1 - y_1 \right) e^{1-t} \end{bmatrix}.$$

*Proof.* The proof of **Proposition A.5** is similar to **Proposition A.4**. One key distinction is the set of differential equations obtained using minimum principle theorem (Basar et al., 2020). We begin with the Hamiltonian:

$$H(\mathbf{x}_t, \mathbf{p}_t, \mathbf{u}_t, t) = \ell(\mathbf{x}_t, \mathbf{u}_t, t) + \mathbf{p}_t^T \left( \mathbf{u}_t + \mathbf{x}_t \right)$$

$$= \frac{1}{2} \|\mathbf{u}_t\|^2 + \mathbf{p}_t^T \mathbf{u}_t + \mathbf{p}_t^T \mathbf{x}_t,$$

which gives us the minimizer of the Hamiltonian $\mathbf{u}_t^* = -\mathbf{p}_t$ and its value function becomes $V^* = \min_{\mathbf{u}_t} H(\mathbf{u}_t, \mathbf{p}_t, \mathbf{u}_t, t) = H(\mathbf{u}_t, \mathbf{p}_t, \mathbf{u}_t^*, t) = -\frac{1}{2}\|\mathbf{p}_t\|^2 + \mathbf{p}_t^T \mathbf{x}_t$. By the minimum principle theorem (Basar et al., 2020),

$$\dot{\mathbf{x}}_t := \frac{\mathrm{d}\mathbf{x}_t}{\mathrm{d}t} = \nabla_{\mathbf{p}} H(\mathbf{x}_t, \mathbf{p}_t, \mathbf{u}_t^*, t) = -\mathbf{p}_t + \mathbf{x}_t; \tag{23}$$

$$\dot{\mathbf{p}}_t := \frac{\mathrm{d}\mathbf{p}_t}{\mathrm{d}t} = -\nabla_{\mathbf{x}} H(\mathbf{x}_t, \mathbf{p}_t, \mathbf{u}_t^*, t) = -\mathbf{p}_t; \tag{24}$$

$$\mathbf{x}_{t_0} = x_0; \tag{25}$$

$$\mathbf{p}_{t_N} = \nabla_{\mathbf{x}} h(\mathbf{x}_{t_N}, t_N) = \gamma A^T (A\mathbf{x}_{t_N} - y_1). \tag{26}$$

This leads to a coupled system of differential equations with boundary conditions as given below:

$$\begin{bmatrix} \dot{\mathbf{x}}_t \\ \dot{\mathbf{p}}_t \end{bmatrix} = \begin{bmatrix} 1 & -1 \\ 0 & -1 \end{bmatrix} \begin{bmatrix} \mathbf{x}_t \\ \mathbf{p}_t \end{bmatrix};$$

$$\mathbf{x}_{t_0} = x_0;$$

$$\mathbf{p}_1 = \gamma A^T (A\mathbf{x}_1 - y_1).$$

This can be solved numerically using ODE solvers, see (Fleming & Rishel, 2012; Basar et al., 2020) for details. Denote $\dot{\mathbf{q}}_t = \begin{bmatrix} \dot{\mathbf{x}}_t \\ \dot{\mathbf{p}}_t \end{bmatrix}$ and $\mathrm{M} = \begin{bmatrix} 1 & -1 \\ 0 & -1 \end{bmatrix}$. We seek a solution of the form $\mathbf{q}(t) = \mathbf{q}e^{\lambda t}$. If $\mathbf{q}(t)$ is a solution of the above problem, then it must satisfy the following eigen value problem:

$$\mathbf{q}e^{\lambda t}\lambda = \mathrm{M}\mathbf{q}e^{\lambda t}. \tag{27}$$

Writing the characteristic polynomial of (27), we get $\det(\mathrm{M} - \lambda \mathrm{I}) = 0$, which gives the eigen values $\lambda = \{1, -1\}$. Substituting these eigen values, we have

$$\begin{bmatrix} 0 & -1 \\ 0 & -2 \end{bmatrix} \begin{bmatrix} q_1 \\ q_2 \end{bmatrix} = \mathbf{0}, \qquad \begin{bmatrix} 2 & -1 \\ 0 & 0 \end{bmatrix} \begin{bmatrix} q_1 \\ q_2 \end{bmatrix} = \mathbf{0},$$

which gives two fundamental solutions. By combining these two, we obtain the final solution

$$\begin{bmatrix} \mathbf{x}_t \\ \mathbf{p}_t \end{bmatrix} = \omega \begin{bmatrix} 1 \\ 0 \end{bmatrix} e^t + \xi \begin{bmatrix} 1 \\ 2 \end{bmatrix} e^{-t},$$

where $\omega$ and $\xi$ can be found using the boundary conditions. Since $\mathbf{x}_0 = x_0$ and $\mathbf{p}_1 = \gamma A^T (A\mathbf{x}_1 - y_1)$, we get $\omega = x_0 - \frac{\gamma}{2} A^T (A\mathbf{x}_1 - y_1) e$ and $\xi = \frac{\gamma}{2} A^T (A\mathbf{x}_1 - y_1) e$. Substituting the values of $\omega$ and $\xi$, we arrive at

$$\begin{bmatrix} \mathbf{x}_t \\ \mathbf{p}_t \end{bmatrix} = \begin{bmatrix} x_0 e^t - \frac{\gamma}{2} A^T (A\mathbf{x}_1 - y_1) e^{1+t} + \frac{\gamma}{2} A^T (A\mathbf{x}_1 - y_1) e^{1-t} \\ \gamma A^T (A\mathbf{x}_1 - y_1) e^{1-t} \end{bmatrix}.$$

This completes the proof of the proposition.

$\square$

**Summary:** Though Appendices A.1-A.3, we have seen the connection between optimal control and diffusion based generation with a personalized terminal constraint. The general strategy has been to derive the optimal controller with known terminal state, and then replace the terminal state in the controller with its estimate using Tweedie's formula. While the controllers so far have an explicit form, in practice, the data distribution is not Gaussian, and thus, we do not have a closed-form expression for the drift of the controller.

This line of analysis, however, points to our method RB-Modulation. As discussed in §4, we incorporate a contrastive style descriptor in our controller's terminal cost and numerically evaluate the drift of the controller at each reverse time step either through back propagation through the score network, or an approximation based on proximal gradient updates.

# B  ADDITIONAL EXPERIMENTS

In this section, we provide implementation details and additional experimental evaluation which have been omitted from the main draft due to limited space.

## B.1 IMPLEMENTATION DETAILS

**Baselines:** We demonstrate the applicability of our method RB-Modulation with StableCascade (Pernias et al., 2024) (released before April 2024). To our best knowledge, RB-Modulation is the first framework that introduces new capabilities to StableCascade by incorporating SOC and AFA modules. Since there are no existing training-free personalization baselines designed for StableCascade, we seek alternatives built on other comparable state-of-the-art models such as SDXL (Podell et al., 2023) and Muse (Chang et al., 2023)[3].

Among alternate training-free baselines, InstantStyle (Wang et al., 2024a) does not directly apply to StableCascade because it requires feature injection into specific layers of an IP-Adapter, which is not available for StableCascade. Similarly, StyleAligned (Hertz et al., 2023) relies on DDIM inversion, which is currently applicable only to single-stage diffusion models. In contrast, StableCascade utilizes a two-stage diffusion process, making the application of standard DDIM inversion (Song et al., 2021a) infeasible. We run the official source code for InstantStyle[4] and StyleAligned[5]. In the absence of a style description, we use "image in style" for DDIM inversion in StyleAligned. Following InstantStyle (Wang et al., 2024a), we also compare with IP-Adapter. We include the quantitative comparison in Table 2, and only compare qualitatively with stronger baselines in Figure 2.

For completeness, we also compare with training-based baselines: StyleDrop (Sohn et al., 2023) and ZipLoRA (Shah et al., 2023). Since the official codebase for StyleDrop[6] and ZipLoRA[7] are not publicly available, we use the third-party implementation and follow the training details in the corresponding papers. It takes 5 minutes for training StyleDrop for 1000 steps and 20 minutes for training each LoRA for ZipLoRA. We train each LoRA with only one reference image for both content and styles to make a fair comparison with other methods. Similarly, we train StyleDrop with only one reference image. When a style description is not provided, we follow the original paper (Sohn et al., 2023) and use "in a [v*] style" instead.

**Tunable parameters.** Our method introduces only two hyper-parameters: stepsize $\eta$ and optimization steps $M$ in **Algorithm 1**. We use DDIM sampling with $\eta = 0.1$ and $M = 3$ for all the experiments. Figure 5 illustrates an overall pipeline of RB-Modulation.

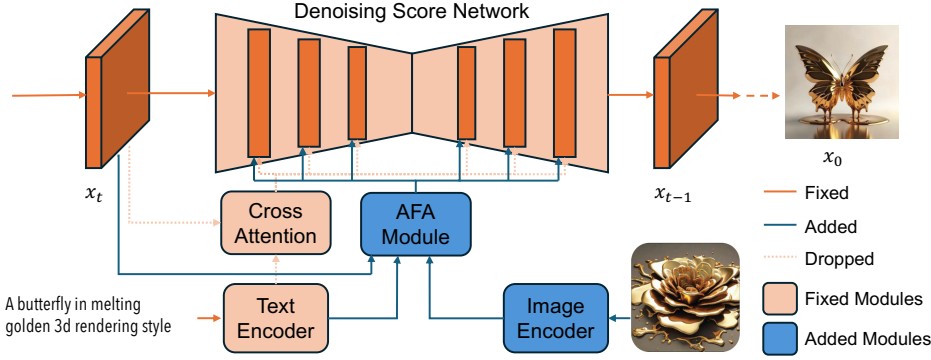

Figure 5: Overall pipeline of **RB-Modulation**. AFA module replaces the cross-attention processor in the denoising UNet, disentangling the content and style of the reference image using CSD [43].

**Content-style composition.** The prompt-guided content-style composition task introduces a new layer of complexity beyond stylization. This task necessitates the disentanglement of the text prompt, reference style image, and reference content image through additional conditioning (Shah et al., 2023; Wang et al., 2024c; Huang et al., 2024b; Guo et al., 2024). Such complexity poses significant challenges for DDIM inversion (Song et al., 2021a) and attention caching mechanisms (Hertz et al., 2023) due to the inherent dependencies on multiple reverse paths.

---

[3]Note that StableCascade and SDXL have comparable performance in prompt alignment whereas StableCascade is more efficient due to a highly compressed semantic latent space (Pernias et al., 2024).

[4]https://github.com/InstantStyle/InstantStyle

[5]https://github.com/google/style-aligned

[6]https://github.com/aim-uofa/StyleDrop-PyTorch

[7]https://github.com/mkshing/ziplora-pytorch

Our AFA module effectively addresses these challenges. It manipulates transformer layers to easily incorporate these additional conditions. The content information is integrated in a manner similar to the style information. Specifically, we use a pre-trained ViT-L/14 model to extract content features in the SOC framework and update the latent embeddings concurrently via the AFA module, using an additional set of keys and values illustrated in Figure 6.

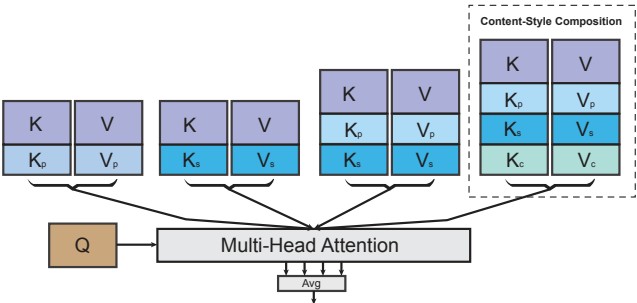

Figure 6: **Attention Feature Aggregation (AFA):** Within the cross-attention layers, the keys and values from the previous layers ($K$,$V$), text embedding ($K_p$,$V_p$), reference style image ($K_s$,$V_s$) and reference content image ($K_c$,$V_c$) are concatenated and processed separately to disentangle the information, which is followed by an averaging layer for the output. $K_c$,$V_c$ and only used for content-style composition.

Furthermore, to better preserve the identity of the foreground content, we extract the desired content using LangSAM[8] based on the content prompt. This step is optional but offers more user control when multiple subjects are present in the reference image.

### B.2 IMPLEMENTATION USING LARGE-SCALE DIFFUSION MODELS

The exact implementation of our control problem (3) is given in **Algorithm 1**, which follows from our theoretical insights. In practice, our controller encounters a challenge when the generative model contains billions of parameters as in StableCascade (Pernias et al., 2024) due to back propagation through the score network, as discussed in §4. Our strategy to overcome this practical challenge involves a proximal gradient update, given in Line 7-8 of **Algorithm 2**. To accelerate the sampling process, we run a few steps ($M = 3$) of gradient descent after initializing $\mathbf{x}_0 = \mathbb{E}\left[X_0^u | X_t^u = \mathbf{x}_t\right]$, resulting in only two hyperparameters to tune: stepsize $\eta$ and the number of optimization steps $M$. Further, since the CSD model expects a clean image to extract style features, we apply the previewer model available in StableCascade on the terminal state before extracting style features. After obtaining the final personalized latent using our **Algorithm 1** and **Algorithm 2**, we follow the decoding process as per the inference pipeline of the adopted generative model. In Table 4, we show the computational overhead of our method in comparison with competing methods.

Table 4: **RB-Modulation** matches the speed of training-free methods and offers 5-20X speedup over training-based methods like StyleDrop [11] and ZipLoRA [10]. For instance, StyleDrop and ZipLoRA require 300 seconds (s) and 1200s, respectively, for training specific components, in addition to their standard inference times of 30s and 40s. RB-Modulation does not use DDIM inversion or additional parameters in the UNet, thus further reducing the computational overhead.

| Method | Runtime (s) | Training-Free | DDIM Inv. | Params in UNet |
|---|---|---|---|---|
| IP-Adapter [21] | 21 | Yes | Yes | Adapters |
| StyleAligned [12] | 39 | Yes | Yes | No |
| InstantStyle [13] | 22 | Yes | Yes | Adapters, ControlNets |
| StyleDrop [11] | 300+30 | No | No | Adapters |
| ZipLoRA [10] | 1200+40 | No | No | 2 LoRAs, 1 Merge layer |
| RB-Modulation (ours) | 44 | Yes | No | No |

---

[8] https://github.com/luca-medeiros/lang-segment-anything

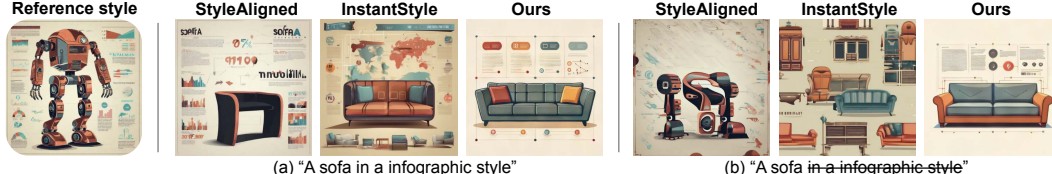

Figure 7: **Impact of style descriptions in the prompt:** (a) When style descriptions are provided, all methods yield better results. (b) Without style descriptions (*e.g.*, hard for users to describe in text), alternative methods could struggle to capture the intended style in the reference image. Our method offers consistent stylization even without explicit style descriptions.

### B.3 IMPACT OF HYPERPARAMETERS ON CONTROLLING STYLE AND CONTENT FEATURES

As detailed in §4 and the ablation study in §6.1, SOC helps disentangle the style and the prompt information by updating the drift field in the standard reverse-SDE. We study the impact of the two hyperparameters present in **Algorithm 1** and **Algorithm 2** that enables this disentanglement, as shown in Figure 8. We found better disentanglement when the step size $\eta = 0.1$ and the number of optimization steps $M = 3$. However, increasing the step size further results in style image information leaking into the output (top row). Additionally, adding more optimization steps increases computational overhead without yielding much performance gain (bottom row).

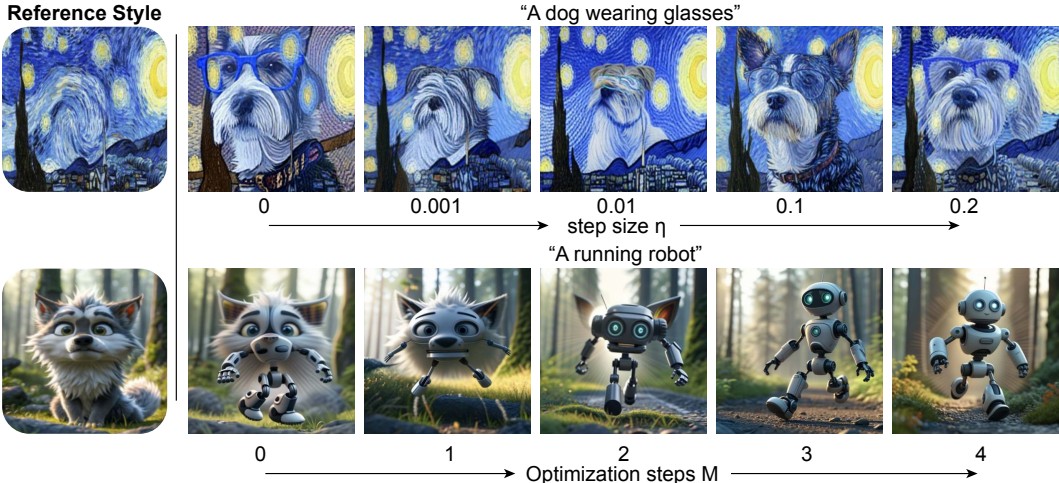

Figure 8: **Qualitative results of different tunable hyperparameters:** Improved style-prompt disentanglement are shown when increasing to our best configurations optimization step size $\eta = 0.1$ and optimization steps $M = 3$.

### B.4 STYLE DESCRIPTION IN TEXT PROMPTS FOR BETTER ASSIMILATION OF UNIQUE STYLES

In addition to the quantitative analysis in §6.1, Figure 7 demonstrates that our method generates consistent stylized results with and without the style description. In contrast, the alternatives fail to accurately follow the prompt when the style description is absent. Although all results show noticeable improvement when the style description is provided, it is often challenging for users to describe styles in many real-world scenarios. We believe our early results by RB-Modulation will pave the way for interesting future research along this direction.

We present additional qualitative results on stylization with (Figure 11) and without (Figure 12) style descriptions using StyleAligned dataset (Hertz et al., 2023). Our results consistently align with the reference style and the prompt, while other methods encounter several issues: (1) difficulty in following prompt guidance, (2) information leakage from the style reference image, and (3) failure

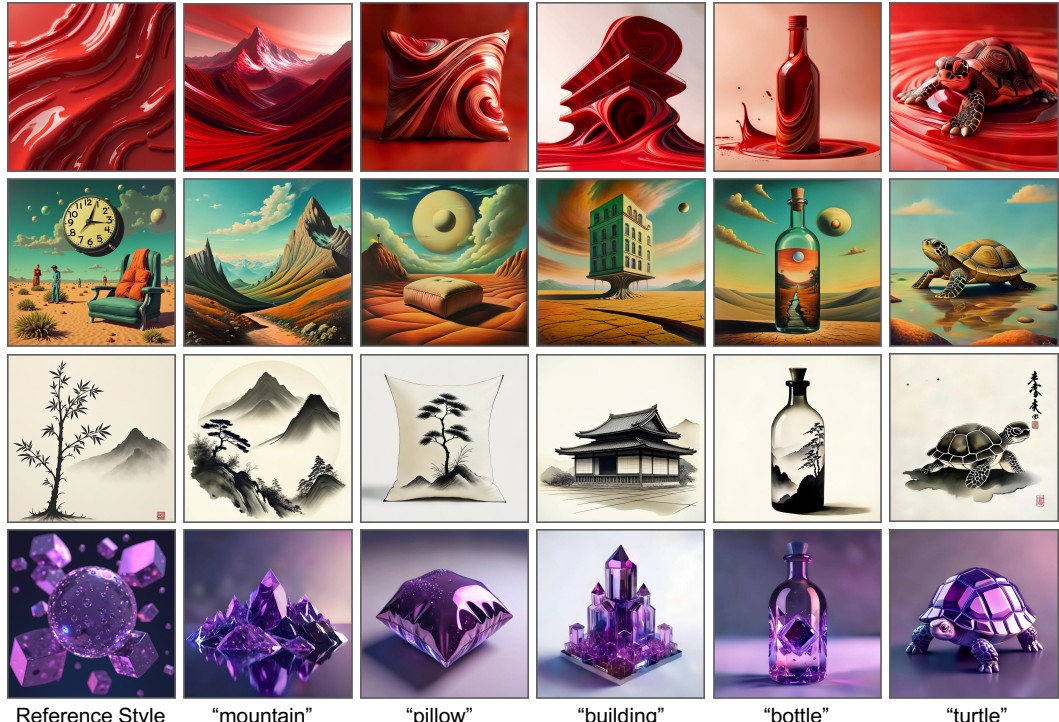

Reference Style     "mountain"     "pillow"     "building"     "bottle"     "turtle"

Figure 9: **A gallery of additional qualitative results on stylization using RB-Modulation.**

to achieve reasonable prompt/style alignment in the absence of style descriptions. Figure 9 presents a gallery of text-driven stylization results using RB-Modulation.

### B.5 EVALUATION CHALLENGES IN MEASURING STYLE AND CONTENT LEAKAGE

In §6, we discussed the limitations of metrics used in previous works (Sohn et al., 2023; Hertz et al., 2023; Shah et al., 2023), such as DINO (Caron et al., 2021) and CLIP-I score (Radford et al., 2021). To quantify these limitations, we use results from our ablation study shown in Figure 3. As illustrated in Figure 10, DINO and CLIP-I scores are not well-suited for measuring style similarity in the presence of content leakage. This is because images with high semantic correlations to the reference style image consistently receive higher scores. For instance, in the top row, although the last two columns visually align more closely with the isometric illustration styles of the reference image, the DirectConcat output featuring a lighthouse receives higher scores. The margin is particularly pronounced for CLIP-I score.

A similar observation can be made in the bottom row, where images containing train-related objects receive higher scores regardless of their stylistic similarity. Conversely, images with less content leakage (as seen in the last column) are assigned lower scores. This indicates that DINO and CLIP-I scores prioritize semantic content over stylistic fidelity, thus failing to accurately measure style similarity in scenarios where content leakage prevails.

On the other hand, our final method (last column), which combines AFA and SOC, demonstrates high scores for both prompt alignment metrics: ImageReward (Xu et al., 2024) and CLIP-T (Radford et al., 2021). This method also shows higher user preference, as evidenced in Table 1. In contrast, the DirectConcat results suffer from information leakage and poor alignment with the prompt, resulting in significantly lower or even negative reward scores.

In the ablation study, our primary focus is on the disentanglement of prompts and reference styles. The conventional metrics fail to accurately reflect true performance due to information leakage. Consequently, we emphasize qualitative demonstrations and place greater importance on user study results, as shown in Table 1, similar to previous approaches (Hertz et al., 2023; Sohn et al., 2023).

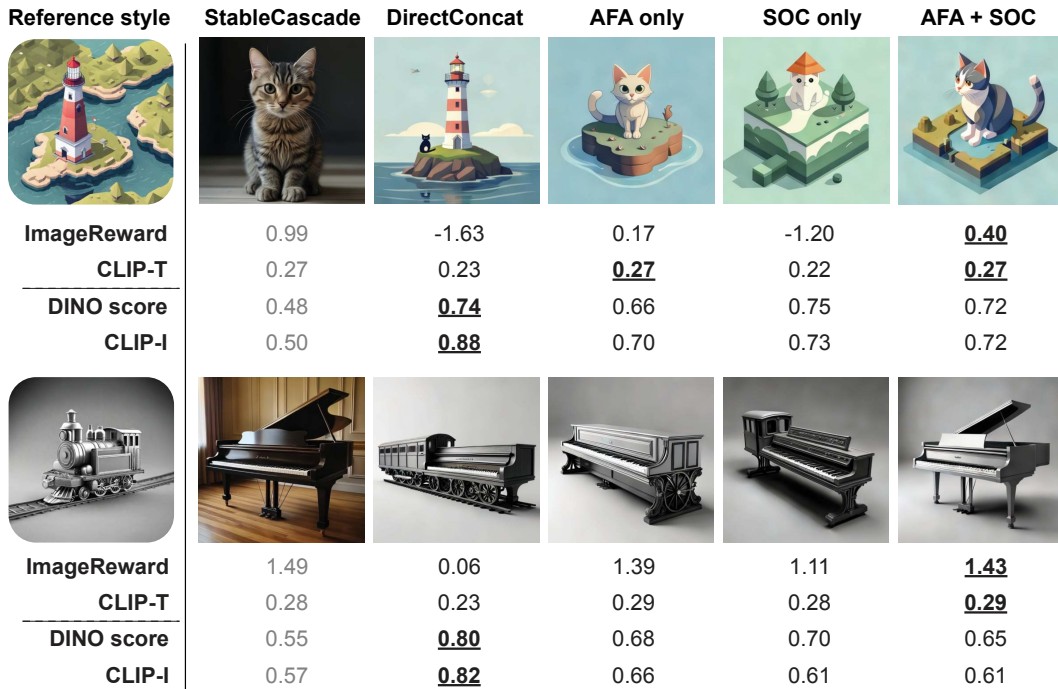

| | Reference style | StableCascade | DirectConcat | AFA only | SOC only | AFA + SOC |
|---|---|---|---|---|---|---|
| **ImageReward** | | 0.99 | -1.63 | 0.17 | -1.20 | **0.40** |
| **CLIP-T** | | 0.27 | 0.23 | **0.27** | 0.22 | **0.27** |
| **DINO score** | | 0.48 | **0.74** | 0.66 | 0.75 | 0.72 |
| **CLIP-I** | | 0.50 | **0.88** | 0.70 | 0.73 | 0.72 |
| **ImageReward** | | 1.49 | 0.06 | 1.39 | 1.11 | **1.43** |
| **CLIP-T** | | 0.28 | 0.23 | 0.29 | 0.28 | **0.29** |
| **DINO score** | | 0.55 | **0.80** | 0.68 | 0.70 | 0.65 |
| **CLIP-I** | | 0.57 | **0.82** | 0.66 | 0.61 | 0.61 |

Figure 10: **Comparison of different evaluation metrics:** The StableCascade output is provided for reference because it doesn't use the reference style image. The highest score for each metric is marked **bold** with underscore. We compare four metrics: ImageReward and CLIP-T score for prompt alignment, DINO and CLIP-I score for style alignment. The prompt for the top row is "A cat" and for the bottom row is "A piano".

## B.6 MORE QUALITATIVE RESULTS ON STYLIZATION AND CONTENT-STYLE COMPOSITION

We also showcase results on consistent style generation using user defined prompts in Figure 13. Our results with different prompts consistently align with the styles while introducing various scenarios following the prompts. The other methods face challenges like information leakage (*e.g.* hiking boots and the monocular) and monotonous scenes (*e.g.* InstantStyle). Note that the original StyleDrop paper (Sohn et al., 2023) has mentioned its difficulty when training with one image without description. We keep the results for completeness even though they are less satisfying. In Figure 15, we provide additional comparison with training-based and training-free personalization approaches. Figure 14 shows stylization given hand drawn reference style images: plastic crayon[9], pencil sketch[10], and commercial paint[11]. In Figure 16, we show qualitative results obtained by integrating the AFA and SOC modules in SDXL (Podell et al., 2023) pipeline, justifying the plug-and-play nature of RB-Modulation.

**Compatibility with ControlNet.** Our method readily adapts to layout guidance via ControlNet (Zhang et al., 2023), as shown in Figure 17. Since ControlNet enhances the denoising network, the proposed method effectively minimizes the terminal cost associated with the expected terminal state, ensuring that SOC remains practical and effective. Furthermore, the AFA module integrates seamlessly by replacing the default attention processor in the denoising network, maintaining its functionality even when ControlNet is employed.

**Controllability of AFA Module.** Figure 18 demonstrates the precise control provided by the AFA module. The pair $(K_p, V_p)$ is computed using the given prompt (e.g., "a cat") without using text description of the reference style image, and $(K_s, V_s)$ using the style attention head of the CSD feature extractor applied to the reference style image. By gradually increasing the strength of the

---

[9] https://ar.pinterest.com/pin/742953269772065667/
[10] https://www.pinterest.com/pin/509891989063791950/
[11] https://www.pinterest.com/pin/ms-paint-drawing-art--690106342901777263/

style image embedding, our method progressively incorporates features from the reference style image, enabling fine-grained control over stylization.

Figure 19 demonstrates the ability of our method RB-Modulation to generate novel and unseen styles by continuously interpolating between CSD style embedding of two reference style images.

In Figure 21, we demonstrate more qualitative results for content-style composition Figure 22 shows the impact of content image in content-style composition. Figure 23 highlights the robustness of RB-Modulation in capturing content-specific features independently of color.

### B.7 ADDITIONAL RELATED WORK

In this section, we discuss missing related works from the main paper. DiffusionDisentanglement (Wu et al., 2023) relies on VGG 16 for perceptual loss and ViT/B-32 for directional CLIP loss, which is prone to content leakage (Wang et al., 2024a). In contrast, our method injects features exclusively from the style attention head of the fine-tuned CSD-CLIP model, ensuring better content-style disentanglement in the AFA module. Additionally, our approach introduces an optimal controller framework to minimize a terminal cost, offering a richer design space and superior controllability compared to (Wu et al., 2023). Lastly, our method reduces sampling bias by optimizing the controller $u$ in Algorithm 1, unlike (Wu et al., 2023), which can provably fail to sample from the correct posterior.

In FreeDoM (Yu et al., 2023), the conditional guidance term $\nabla_{\mathbf{x}_t} \log p(\cdot|\mathbf{x}_t)$ is approximated by the gradient of an energy function, $\nabla_{\mathbf{x}_t} \mathcal{E}(\cdot, \mathbf{x}_t)$. Our **Algorithm 1** differs by replacing $\nabla_{\mathbf{x}_t} \log p(\cdot|\mathbf{x}_t)$ with a controller $u$, optimized to minimize this approximation error. Algorithm 2 in FreeDom introduces a time-travel resampling strategy to mitigate poor guidance problem in their Algorithm 1 by iteratively noising and denoising the intermediate latents. While effective, this process is computationally expensive. In contrast, our approach (**Algorithm 2**) is grounded in optimal control, where we optimize the expected terminal state to satisfy constraints, such as aligning the style of the generated image with the input. Thus, our **Algorithm 2** avoids the need for gradient computation through the denoising score network, which is particularly expensive for large-scale models like SDXL or StableCascade. Additionally, we propose a novel attention processor, namely AFA module to disentangle content and style, whereas FreeDoM uses the standard attention processor, known to suffer from content leakage (Hertz et al., 2023; Wang et al., 2024a).

PARASOL (Tarrés et al., 2024) and Diff-NST (Ruta et al., 2023) are training-based methods, while our approach is entirely training-free. PARASOL requires supervised data via a cross-modal search (Section 3.1 in (Tarrés et al., 2024)) to train both the denoising U-Net and a projector network. Diff-NST (Ruta et al., 2023) trains the attention processor by targeting the 'V' values within the denoising U-Net architecture. In contrast, our method uses two training-free modules: the AFA module replaces the default attention processor in the denoising U-Net to disentangle content and style, and the SOC module minimizes a terminal cost to enhance stylization and content-style composition.

### B.8 HUMAN EVALUATION TO DISCERN HIGHLY SUBJECTIVE NATURE OF STYLE

We conduct a user study with 155 participants via Amazon Mechanical Turk using 100 styles from the StyleAligned dataset (Hertz et al., 2023). The study requires no personally identifiable information of the participants. There is no risk incurred and no vulnerable population. The standard guidelines have been followed while conducting the user study.

We first provide participants with instructions to familiarize them with the relevant terminologies. For each style, we randomly sample three outputs using three different prompts. Participants see two rows of model outputs in random order (3 images per row) and answer 3 questions, as illustrated in Figure 20.

1. In which row below, the images align better with the reference style image?

2. In which row below, the images align better with the reference text prompt above each image?

3. In which row below, the images overall align better with the reference style image AND the text prompt above each image AND with high quality?

For each question, participants choose one of three options. We collect 8 responses for each question, with each question comparing our method against one of the alternatives. In total, we gathered 7,200 responses.

### B.9 FAILURE CASES OF TRAINING-FREE STYLIZATION USING RB-MODULATION

In Figure 24, we illustrate stylization of different letters using a single reference style image. Although our method captures the intended style and generates prompted letters, we notice that there is an inherent tendency to generate upper-case letters (Figure 24 (a)), even though it is prompted to generate lower-case letters. Upon further investigation, we observed that this issue stems from the underlying generative model StableCascade, as shown in Figure 24 (b). This highlights a crucial limitation of our method. As a training-free method, RB-Modulation shares a concern with other training-free methods (Wang et al., 2024a; Hertz et al., 2023; Jeong et al., 2024) that the performance is influenced by the original generative prior.

### B.10 LIMITATIONS

In this paper, we proposed a framework and demonstrated its efficacy by incorporating a style descriptor (Somepalli et al., 2024) in a pre-trained diffusion model (Pernias et al., 2024). The inherent limitations of the style descriptor or diffusion model might propagate into our framework. We believe these limitations can be addressed by an appropriate descriptor or a generative prior.

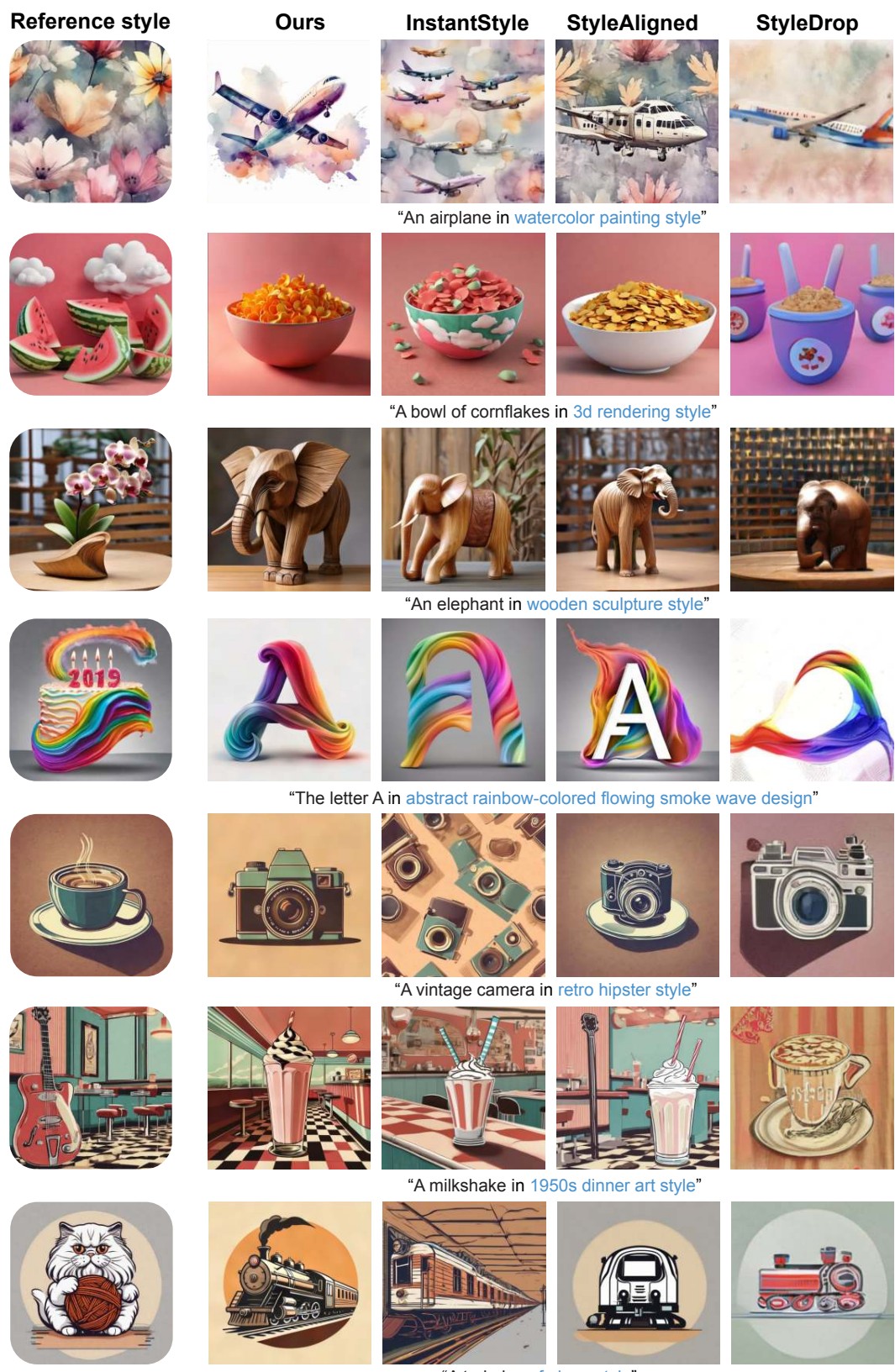

Figure 11: **Additional qualitative results for stylization with style description:** While the alternative methods face challenges like following the prompts (*e.g.*, multiple airplanes instead of an airplane) and information leakage (*e.g.*, the clouds on the cornflake bowl and the guitar in the milkshake image), our method demonstrates strong performance on both prompt and style alignment. Style description is in blue.

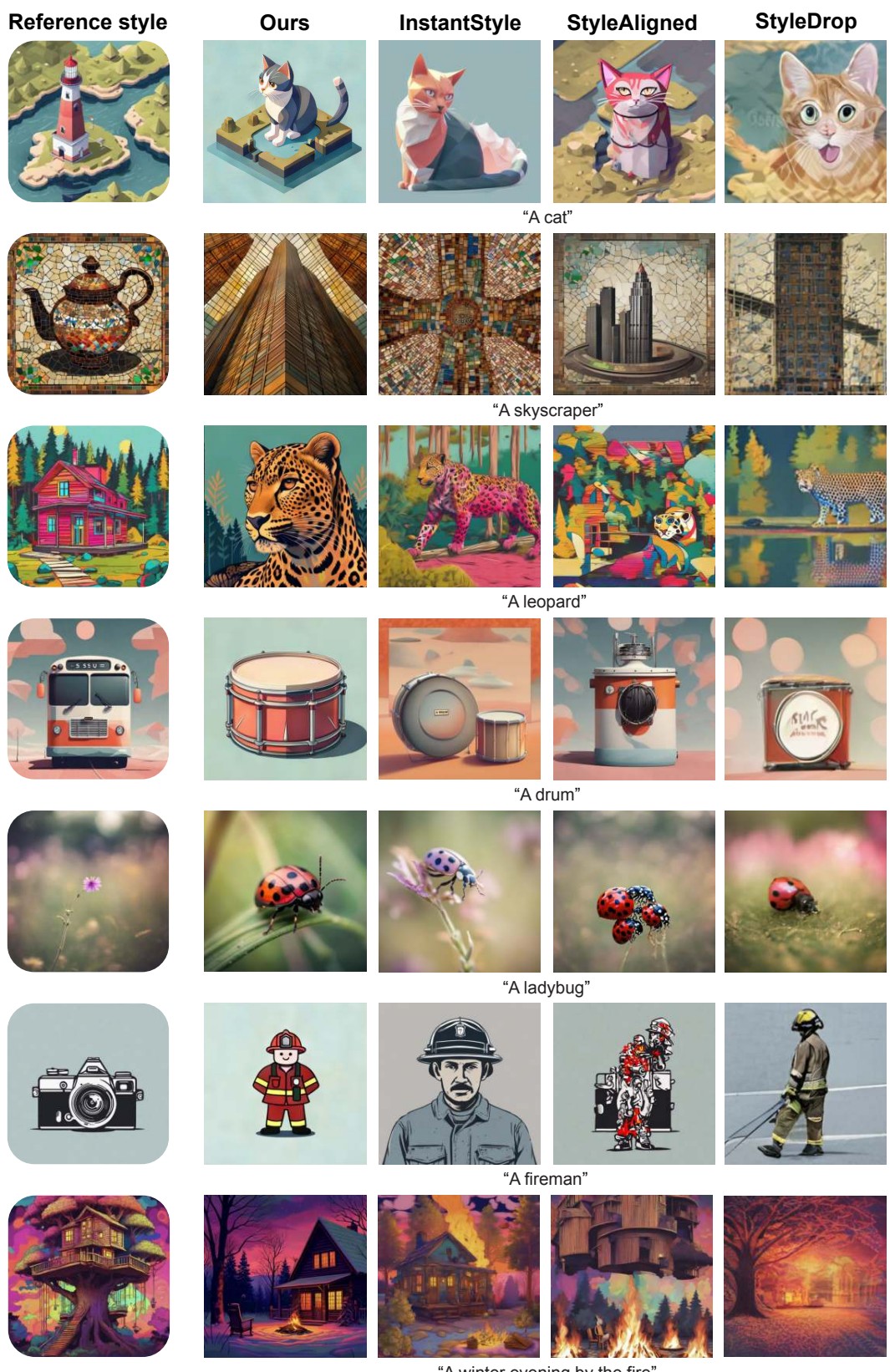

Figure 12: **Additional qualitative results for stylization without style description:** StyleAligned and StyleDrop show severe performance drop after removing the style descriptions (*e.g.*, see fireman and cat images). InstantStyle results show more information leakage (*e.g.*, the pink ladybug and leopard), whereas no obvious performance drop is observed in our results.

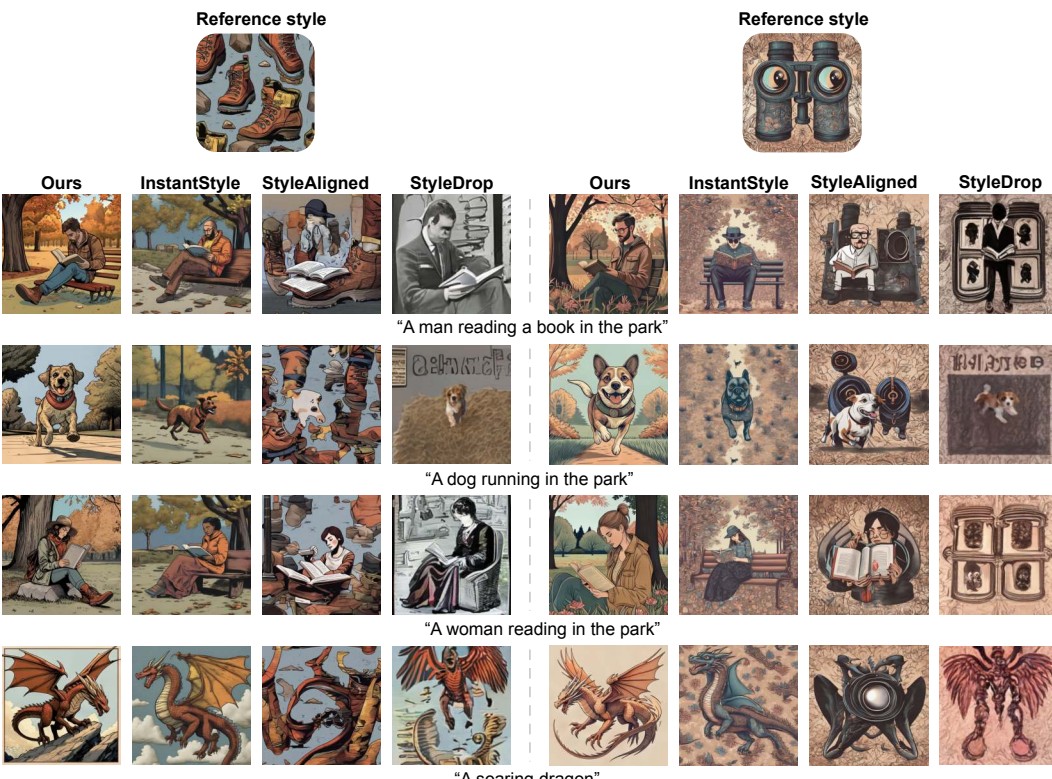

Figure 13: **Additional qualitative results for consistent stylization for user defined prompts:** With no style description, our results demonstrate more diversity while following the styles and prompts. InstantStyle results show monotonous scenes and StyleAligned results suffer from severe information leakage. We report StyleDrop results for completeness and it is known to perform worse with no style description and single training image (Sohn et al., 2023).

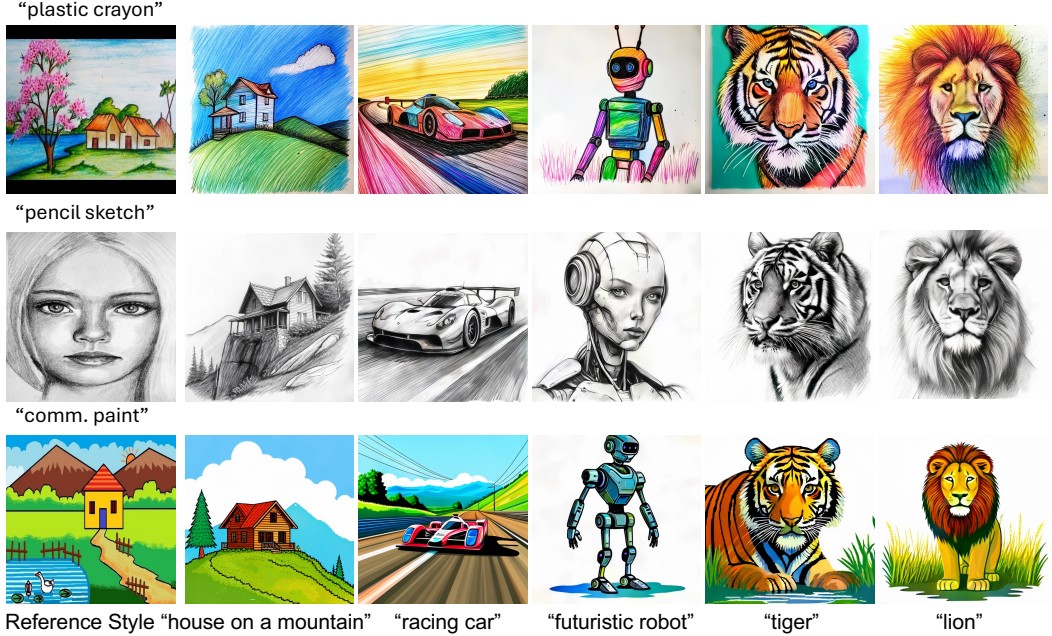

"plastic crayon"

"pencil sketch"

"comm. paint"

Reference Style "house on a mountain" "racing car" "futuristic robot" "tiger" "lion"

Figure 14: **Qualitative results for hand-drawn reference style images.** The proposed method is agnostic to real or generated reference images. Given hand drawn reference style images (e.g., "paint" from a commercial service provider) and desired text prompts (e.g., "a tiger"+style description), RB-Modulation captures the reference style in the generated content image. Please see §B.6 for the reference style image credits.

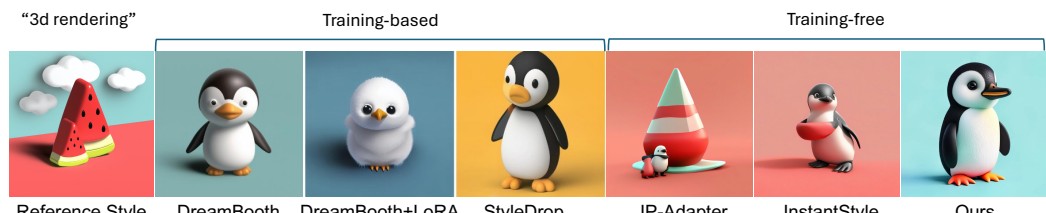

"3d rendering"  Training-based      Training-free

Reference Style DreamBooth DreamBooth+LoRA StyleDrop IP-Adapter InstantStyle Ours

Figure 15: **Qualitative comparison with classical personalization methods.** The proposed method significantly outperforms other training-free methods while remaining comparable to or better than classical training-based personalization approaches. Prompt:"a baby penguin in 3d rendering style."

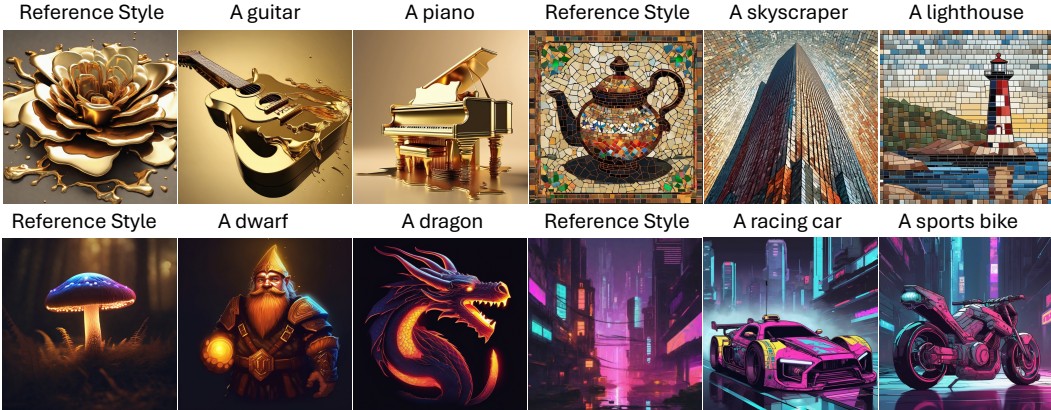

Figure 16: **Qualitative results using SDXL (Podell et al., 2023) as base model.** This verifies the plug-and-play nature of RB-Modulation for training-free personalization.

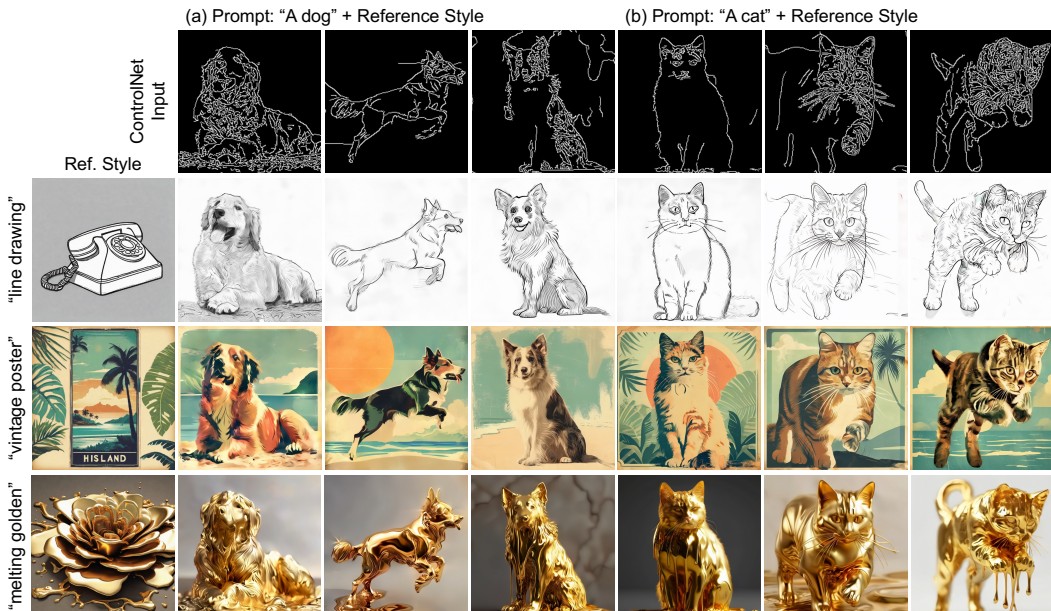

Figure 17: **Qualitative results demonstrating compatibility with ControlNet (Zhang et al., 2023).** Given the Canny edge map of a reference content and an image of a reference style, the proposed method effectively controls the pose of the generated samples while accurately capturing the desired style.

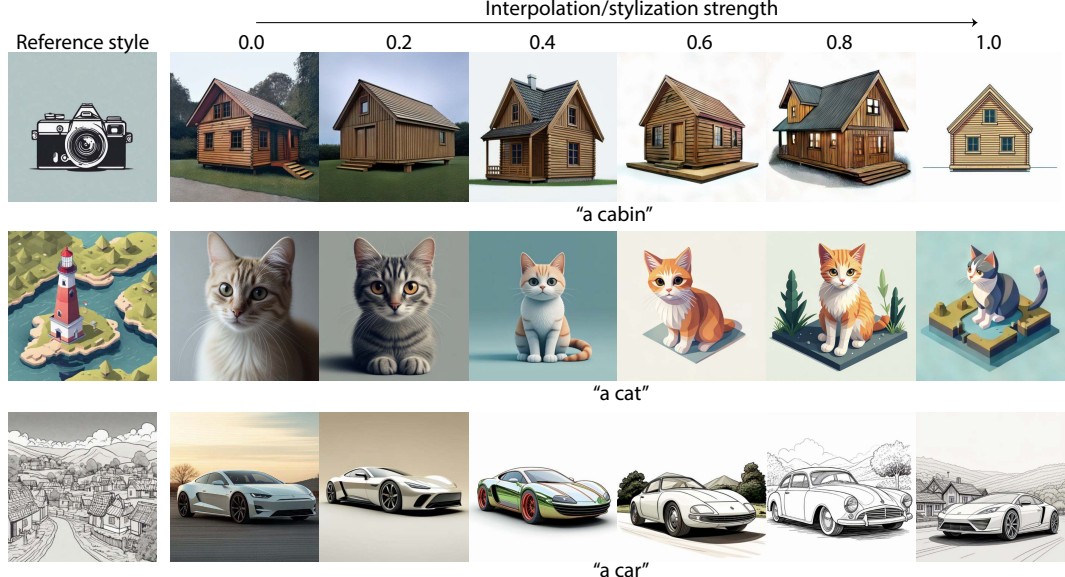

Figure 18: **Qualitative results showing controllability of our method for stylization.** By progressively increasing the strength of the style image embedding derived from the CSD style descriptor, our method gradually integrates features from the reference style image, providing fine-grained control over stylization.

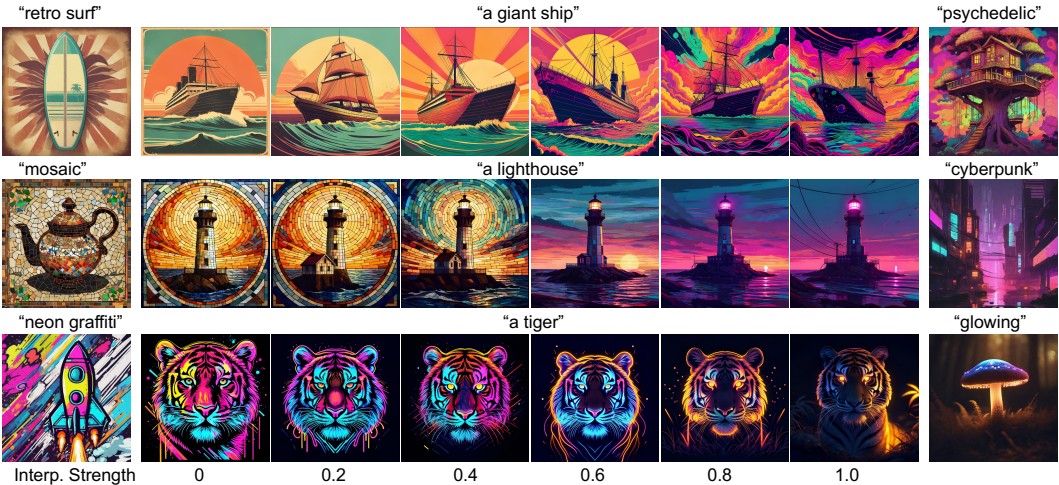

Figure 19: **Qualitative results showing interpolation of two different reference style images.** The interpolation strength parameter provides additional control for blending features from multiple reference styles (e.g., "a lighthouse in mosaic art style" → "a lighthouse in cyberpunk art style"). This highlights RB-Modulation's capability to generate novel and previously unseen styles.

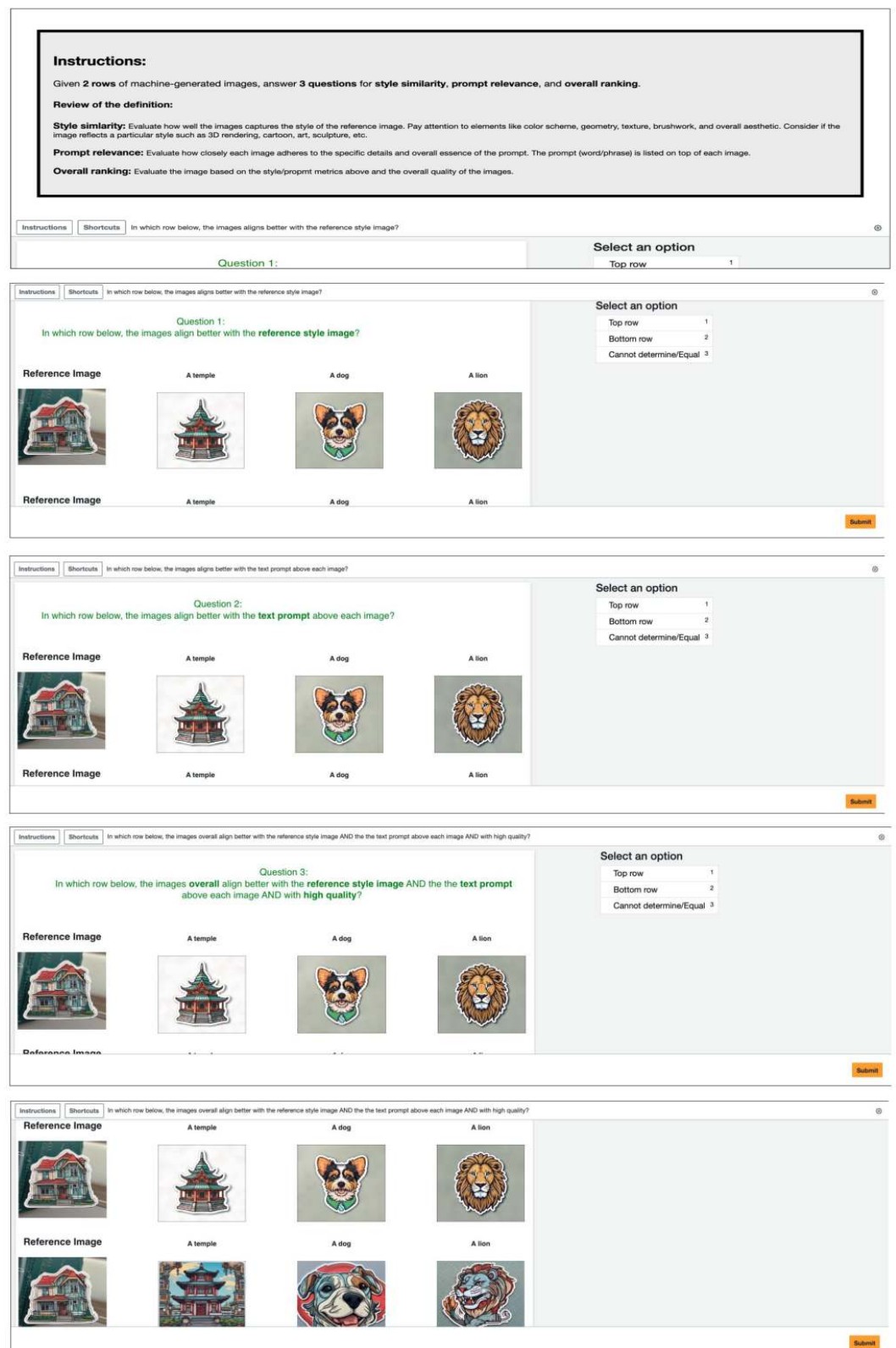

Figure 20: **User study interface:** Three randomly sampled outputs are shown for each method given a style reference image, forming two rows of images. The users are asked to answer three questions on (1) style alignment (2) prompt alignment and (3) overall alignment and quality.

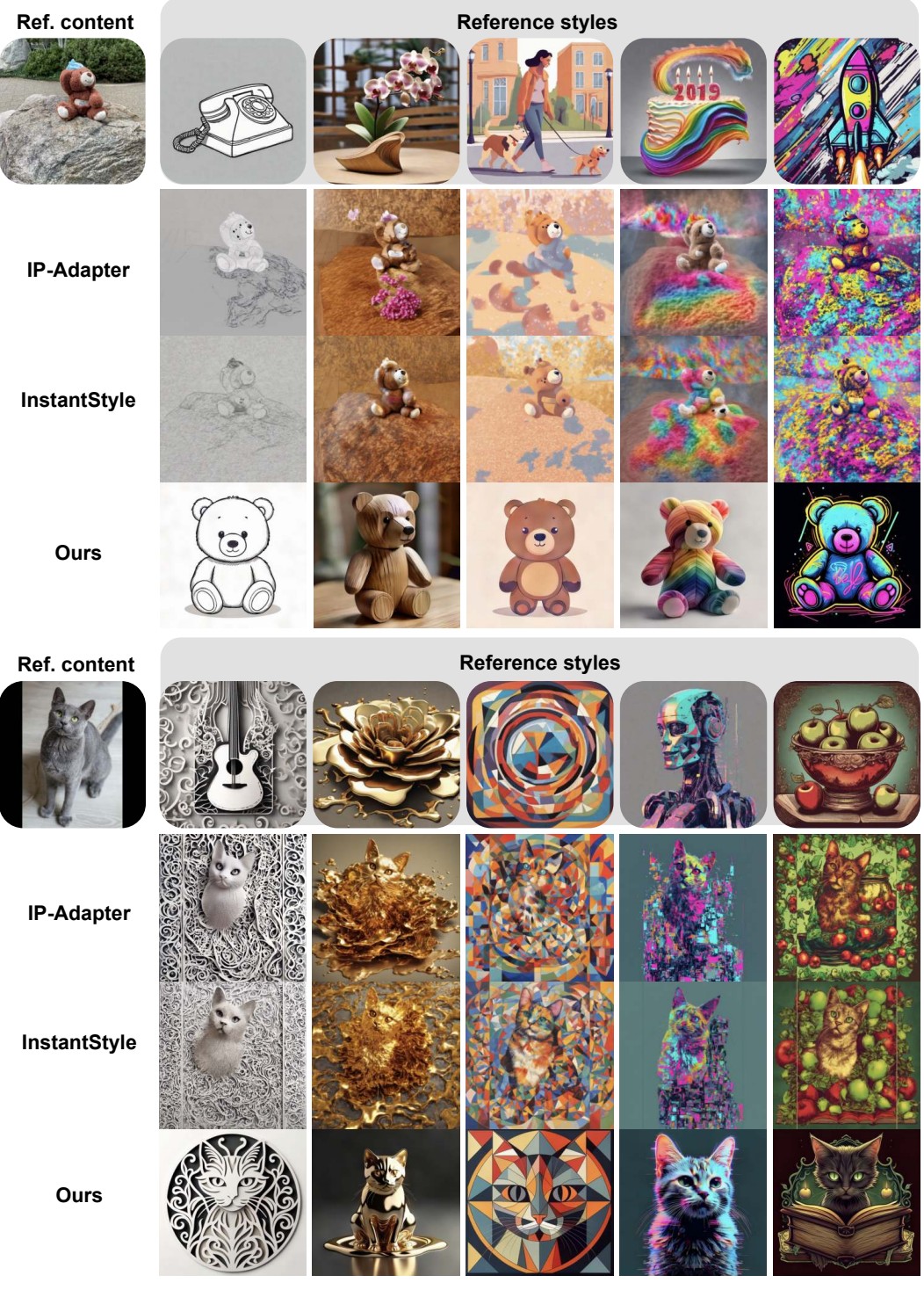

Figure 21: **Additional qualitative results for content-style composition:** Our results show better prompt and style alignment while preserving reference content without leaking contents from the reference style images (*e.g.* background of the first column and fruits in the last column,). Unlike compared baselines, our method is not restricted to a fixed pose of the reference content image, illustrating sample diversity.

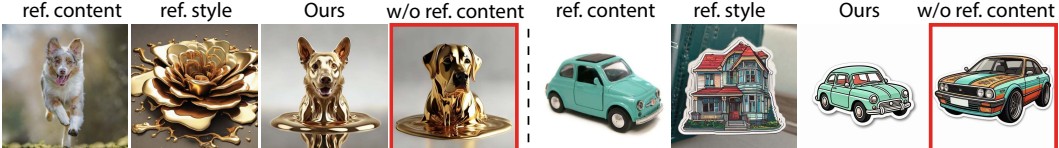

Figure 22: **Qualitative results on content-style composition to illustrate the impact of content image.** Excluding the content reference image (i.e., removing $K_c$ and $V_c$ from the AFA module) results in a loss of content details, such as the dog breed and car type, as highlighted in the red box.

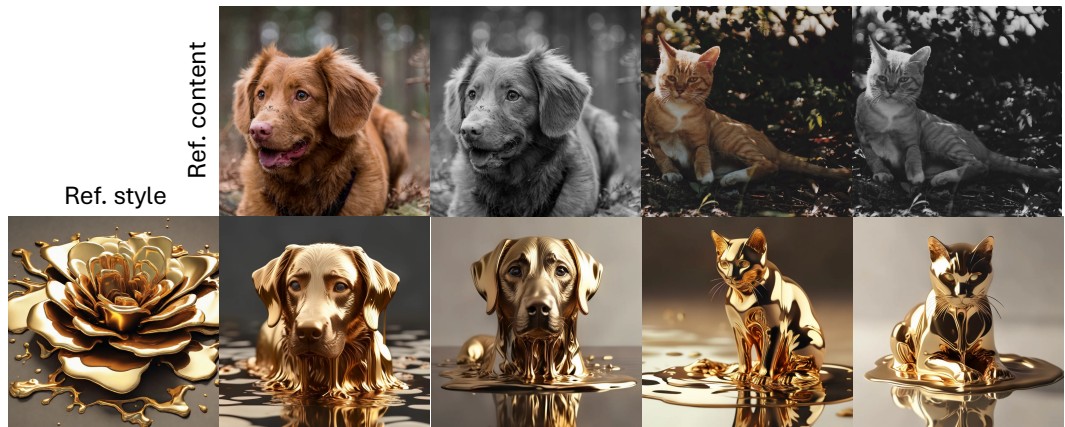

Figure 23: **Qualitative comparisons for content-style composition by graying out the reference content image.** Notably, the content (e.g., dog) is effectively transferred even after the grayscale transformation, demonstrating the robustness of our method in capturing and transferring content-specific features independently of color.

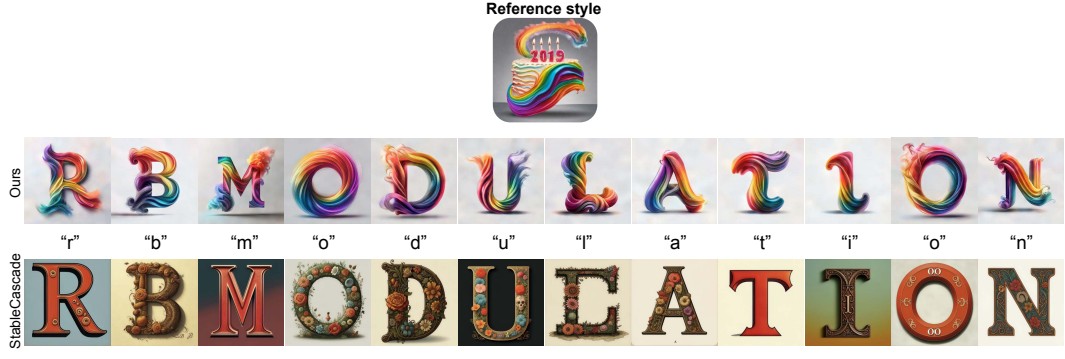

Figure 24: **Failure cases for stylization:** The top row shows the results of our method, RB-Modulation, while the bottom row displays the results of the backbone, StableCascade. Notably, the stylized images do not adhere to the prompt, "lower-case letter". This highlights the limitations imposed by the pre-trained generative priors on the capabilities of training-free personalization models (top row).

