# OpenReview forum: "RB-Modulation: Training-Free Stylization using Reference-Based Modulation"
_ICLR.cc/2025/Conference — ICLR 2025 Oral_

### Official Review · Reviewer_YTQn · 2024-10-28

**Soundness:** 4
**Presentation:** 3
**Contribution:** 3
**Rating:** 8
**Confidence:** 4

**Summary:**

The authors propose RB-Modulation, a method that allows training free stylization and content-style composition with diffusion models. The authors reframe the personalization task as an optimal control problem, and propose an additional Attention Feature Aggregation (AFA) module that performs content/style disentanglement. The authors include extensive qualitative/quantitative results that demonstrate the effectiveness of their proposed method.

**Strengths:**

1. AFAIK, personalization with diffusion models are inherently tricky in the sense that training based methods take a long time or training free methods lack in fidelity or fail in content/style disentanglement. The proposed RB-Modulation takes on a training free method that achieves both content/style disentaglement in a training free manner.

2. Evaluations are comprehensive and the proposed method surpasses previous SOTA methods. Qualitative results are especially impressive.

3. The paper is well written and easy to follow.

**Weaknesses:**

1. The choice of the baseline model (StableCascade) seems like a design choice. Authors might want to provide additional results on more widely adapted baseline models (e.g. SDXL), since this can show whether the inherent high fidelity results stems from the potency of StableCascade or their proposed method.
2. Most of the qualitative results seem to be on generated reference style/content images. The proposed method seems to be agnostic to whether the reference image is generated or not. Results with non-generated reference images would be appreciated.
3. Authors use Consistent Style Descriptor (CSD) to extract style features. Are there alternatives for this module? What was the main reasoning behind this choice?
4. For content style composition, it seems like the content is also processed in the same manner as style. Is this optimal in terms of content identity preservation? In algorithm 1 and 2, it seems like the image latent is only updated by the loss w.r.t the style descriptor loss. Would this not result in information loss about the identity of the content?
5. In figure 1, the resulting images do not seem to hold the content identity of the reference content image, but instead seems to heavily rely on the given text description. Using examples from figure 2, additional content-style composition results on including/excluding $K_c, V_c$ would be appreciated.

**Questions:**

1. Does RB-Modulation also work for recent flow based models? Does the standard reverse SDE for the OU process hold for flow based models, or will the optimal controller have to be redefined?

---

> ### Author Response · Authors · 2024-11-18
> **Official Response by Authors**
>
> Dear Reviewer YTQn,
>
> Thank you for recognizing that our method achieves **content/style disentanglement without training, with comprehensive evaluations, state-of-the-art performance, and impressive qualitative results**. We are glad that you found the paper **well-written and easy to follow**. Below, we address your remaining questions.
>
> **(Q1) The choice of the baseline model (StableCascade) seems like a design choice. Authors might want to provide additional results on more widely adapted baseline models (e.g. SDXL), since this can show whether the inherent high fidelity results stems from the potency of StableCascade or their proposed method.**
>
> (A1) We have now included Figure 16 to demonstrate that (1) the plug-and-play nature of RB-Modulation and (2) the inherent high fidelity results indeed stem from our algorithmic innovation.
>
> **(Q2) Most of the qualitative results seem to be on generated reference style/content images. The proposed method seems to be agnostic to whether the reference image is generated or not. Results with non-generated reference images would be appreciated.**
>
> (A2) To ensure a fair comparison, we evaluated our method on the same dataset used in prior work StyleAligned. Following the reviewer’s suggestion, we have now included non-generated reference images in Figure 14.
>
> **(Q3) Authors use Consistent Style Descriptor (CSD) to extract style features. Are there alternatives for this module? What was the main reasoning behind this choice?**
>
>
> (A3) RB-Modulation is a generic framework that can incorporate various feature extractors in its terminal cost. For instance, neural networks could be used to extract specific features like lighting, shadow, or color. For style transfer, extractors such as ALADIN-ViT or ALADIN-NST [1] can also be used. We chose CSD due to its large-scale training on 511,921 images and 3,840 style tags, covering a wide variety of practical styles.
>
>
> **(Q4) For content style composition, it seems like the content is also processed in the same manner as style. Is this optimal in terms of content identity preservation? In algorithm 1 and 2, it seems like the image latent is only updated by the loss w.r.t the style descriptor loss. Would this not result in information loss about the identity of the content?**
>
> (A4) For content-style composition, the content is processed differently from the style, as discussed in L264-269.
>
> For clarity, we summarize the key points from that section here. The pre-trained ViT in CSD includes two attention heads—one for style features and one for content features—which can be selected based on the application. We use the content attention head to extract content-specific features and the style attention head to extract style-specific features. In Algorithms 1 and 2, we simplify notation by using $\Psi(\cdot)$ as a general feature extractor; it represents a style feature extractor for stylization and a combination of content and style extractors for content-style composition.
>
> **(Q5) In figure 1, the resulting images do not seem to hold the content identity of the reference content image, but instead seems to heavily rely on the given text description. Using examples from figure 2, additional content-style composition results on including/excluding Kc,Vc would be appreciated.**
>
> (A5) The base model is a text-to-image generative model, so it naturally relies heavily on the given text description, which typically provides coarse-grained control. The AFA and SOC modules are designed to add fine-grained control beyond the base model's capabilities. To clarify this, we have now included experiments with and without $K_c, V_c$ in Figure 21 as suggested by the reviewer.
>
>
> **(Q6) Does RB-Modulation also work for recent flow based models? Does the standard reverse SDE for the OU process hold for flow based models, or will the optimal controller have to be redefined?**
>
> (A6) Yes, RB-Modulation can work for recent flow-based models, which is an excellent direction for future research. Here are some preliminary thoughts:
> The standard reverse SDE for the OU process doesn’t hold directly for flow models, as the forward process in flows is not governed by an OU process; therefore, its time-reversal differs from the standard reverse SDE of the OU process. Instead, the forward process in flows is governed by a different SDE with different drift and diffusion coefficients, leading to a distinct reverse SDE. Importantly, the theoretical tools used in Appendix A.2 and Appendix A.3 hold for general SDEs, and therefore can be used to rederive  the optimal controller for this new SDE. Thank you for the insightful question.
>
> ### Reference
>
> [1] Ruta, Dan, Gemma Canet Tarres, Alexander Black, Andrew Gilbert, and John Collomosse. "Aladin-nst: Self-supervised disentangled representation learning of artistic style through neural style transfer." arXiv preprint arXiv:2304.05755 (2023).

---

> > ### Comment · Reviewer_YTQn · 2024-11-22
> >
> > Thanks for addressing my comments. The revised draft looks good to me and the additional experiments addressed all my questions/concerns. I would like to keep my rating.

---

> > > ### Author Response · Authors · 2024-11-22
> > > **Official Comment by Authors**
> > >
> > > Thank you for your prompt response. We are pleased to hear that the additional experiments in the revised draft addressed all your questions/concerns.

---

### Official Review · Reviewer_iCFz · 2024-11-02

**Soundness:** 3
**Presentation:** 3
**Contribution:** 3
**Rating:** 6
**Confidence:** 4

**Summary:**

The paper presents am encoder-based (i.e. fine-tuning free) method for style personalization/ customization in text2image diffusion models.  The approach (RB-Modulation) is to consider the reverse dynamics of diffusion as a stochastic control problem with a terminal cost objective that aligns extracted style features with the exemplar style features using a pre-existing  style descriptor [Somepalli, CVPR 2024].   A novelty of the approach is the ‘AFA module’ proposed which separates style and content using cross-attention to mitigate content leakage from the reference image.  The authors claim but do not comprehensively show this allows controlled integration of style and content elements.

**Strengths:**

The formulation of the feed-forward instance personalization problem using a ‘optimal controller’ is novel over the majority of prior works in this category that focus up adaptation.  However the practicalities of the problem seem to demand iterative reverse/feed-forward steps which negates a lot of the benefit of theoretical arguments and underlying assumptions presented.

**Weaknesses:**

Deeper comparisons could have been made to the NST literature which bear some relevance both in task and in approach particularly the recent diffusion based approaches.  Specifically the paper bears some similarity with prior works that partly reverse the diffusion process and then run it ‘forward’ again with style conditioning either from CLIP or from style descriptors.  For example ‘Uncovering the Disentanglement Capability in Text-to-Image Diffusion Models’ [Wu et CVPR 2023) and PARASOL [Canet-Tarres et al., CVPR 2024] which uses ALADIN style descriptors for conditioning as does DIFF-NST [Ruta et al., ECCVW 2024].  Although the literature survey is quite broad in the areas of NST and style conditioned diffusion, there is limited focus on prior descriptor-based stylization work including these works and how the proposed approach fundamentally differs especially considering the use of cross-attention in AFA versus these works.

The paper claims extensive experiments covering stylization and content-style composition as one of their three contributions.  It would have been helpful to see experiments exhibiting more nuanced control over style i.e. fine-grained variations of similar style, to exercise the controllability offered via the style descriptor conditioning.  Conditioning on text is sufficient to give coarse grain control over style (e.g. neon may be specified versus a descriptor for neon, as shown in the examples).  The fine-grained control offered by a continuous style descriptors in a feed-forward framework seems the main benefit but is not discussed or explored.  Similarly a matrix-like experiment showing descriptor interpolation and varying weights on the disentangled content/style would have been helpful to show the practical use/controllability of the approach.

**Questions:**

Please see the above questions in the weakness section.  Overall this appears a technically sound paper but that does not fully contextualize its contribution in the literature or fully evidence its practicality.  The authors can consider addressing this in the rebuttal.

POST-REBUTTAL:  Additional works raised in this review have been discussed within Section 2 of the main paper to contextualize the contribution.  Additional experiments have been added to the supplemental material to show the visual results of continuous variation of style strength and of style interpolation.  I have raised my initial score accordingly to reflect a recommendation to accept the paper.

---

> ### Author Response · Authors · 2024-11-18
> **Official Response 1/2 by Authors**
>
> Dear Reviewer iCFz,
>
> Thank you for acknowledging that **our optimal controller formulation is novel over the majority of prior works in this category.** Below, we address your remaining questions.
>
> **(Q1) A novelty of the approach is the ‘AFA module’ proposed which separates style and content using cross-attention to mitigate content leakage from the reference image. The authors claim but do not comprehensively show this allows controlled integration of style and content elements.**
>
> (A1) We provide analysis in Figures 8 and 10 to demonstrate the controlled integration of style while effectively separating content from the reference style image. Additionally, Figures 11 and 12 offer further qualitative results that highlight content leakage. We have added a new Figure 18 to show controlled integration of style as suggested by the reviewer. We also included Figure 17 to demonstrate control of structural elements via ControlNet.
>
> **(Q2) However the practicalities of the problem seem to demand iterative reverse/feed-forward steps which negates a lot of the benefit of theoretical arguments and underlying assumptions presented.**
>
> (A2) We believe there may have been a misunderstanding regarding this key aspect of our paper. The iterative refinement only happens in the reverse process and there are no feed-forward steps. This retains the benefits because (1) it is performed in the latent space (24x24), where gradient descent adds minimal inference time, as demonstrated in Table 4; and (2) our proximal Algorithm 2 avoids the need for back-propagation through the score network, as outlined in Section 4 (Lines 212-229). Overall, RB-Modulation matches the efficiency of training-free methods and provides a 5-20X speedup over training-based approaches such as StyleDrop [11] and ZipLoRA [10]. We hope this clarifies the efficiency and practical benefits of our approach.
>
> **(Q3) Deeper comparisons could have been made to the NST literature which bear some relevance both in task and in approach particularly the recent diffusion based approaches. How the proposed approach fundamentally differs especially considering the use of cross-attention in AFA versus these works.**
>
> (A3) The proposed method is fundamentally different from these approaches [1,2,3]. We have added these related works in the revised paper (L1302-1318).
>
> DiffusionDisentanglement [1] relies on VGG 16 for perceptual loss and ViT/B-32 for directional CLIP loss, which is prone to content leakage [4]. In contrast, our method injects features exclusively from the style attention head of the fine-tuned CSD-CLIP model, ensuring better content-style disentanglement in the AFA module. Additionally, our approach introduces an optimal controller framework to minimize a terminal cost, offering a richer design space and superior controllability compared to [1]. Lastly, our method reduces sampling bias by optimizing the controller $u$ in Algorithm 1, unlike [1], which can provably fail to sample from the correct posterior [5].
>
> PARASOL [2] and Diff-NST [3] are training-based methods, while our approach is entirely training-free. PARASOL [2] requires supervised data via a multi-modal search (Section 3.1 in [2]) to train both the denoising U-Net and a projector network. Diff-NST [3] trains the attention processor by targeting the 'V' values within the denoising U-Net architecture. In contrast, our method uses two training-free modules: the AFA module replaces the default attention processor in the denoising U-Net to disentangle content and style, and the SOC module minimizes a terminal cost to enhance stylization and content-style composition.

---

> > ### Author Response · Authors · 2024-11-18
> > **Official Response 2/2 by Authors**
> >
> > **(Q4) The paper claims extensive experiments covering stylization and content-style composition as one of their three contributions. It would have been helpful to see experiments exhibiting more nuanced control over style i.e. fine-grained variations of similar style, to exercise the controllability offered via the style descriptor conditioning.**
> >
> > (A4) We respectfully note that we evaluate our method using the same dataset as prior state-of-the-art approaches. Compared to the baselines, we have **extensive experiments** on stylization with and without text description of style, and have collected 7200 human evaluations along with quantitative results. As requested by reviewers, we have now added stylization based on hand-drawn  styles (Figure 14), comparison with classical personalization methods (Figure 15), demonstrated compatibility with another popular base model SDXL (Figure 16), showed layout guidance via ControlNet (Figure 17), and illustrated controllability using AFA module (Figure 18). Regarding content-style composition, we demonstrate the effectiveness of our method in following prompts in Figure 4, quantitative results in Table 3, and additional composition results in Figure 20. As requested by reviewers, we have now included Figures 21 and 22 to show the impact of content image in composition.
> >
> > Fine-grained control over similar styles primarily tests the capabilities of the style descriptor itself, which is beyond the primary focus of our contribution. For an in-depth analysis of the descriptor, we kindly invite the reviewer to refer to the CSD paper. Our main objective is to present a versatile framework that can integrate any off-the-shelf descriptor. If CSD is found to be less effective in certain applications, it can be easily replaced with a more suitable descriptor, as discussed in our limitations (L535-538).
> >
> > **(Q5)  Conditioning on text is sufficient to give coarse grain control over style (e.g. neon may be specified versus a descriptor for neon, as shown in the examples). The fine-grained control offered by a continuous style descriptors in a feed-forward framework seems the main benefit but is not discussed or explored. Similarly a matrix-like experiment showing descriptor interpolation and varying weights on the disentangled content/style would have been helpful to show the practical use/controllability of the approach.**
> >
> > (A5) We discuss the fine-grained control provided by the style descriptor through our SOC module in L460-466 of the main paper, as well as in Figure 8 (Appendix B.3) and Figure 10 (Appendix B.5) in appendix.
> > Describing a style image via text is often challenging (L26-36), making text-guided coarse-grained control difficult. Even in these cases, our method effectively captures style directly from the reference image, as shown in Figure 12.
> > We have added a new experiment in Figure 18 to highlight the fine-grained control enabled by the continuous style descriptor in the AFA module. By progressively increasing the strength of the style image embedding, our method gradually integrates features from the reference style image, providing fine-grained control over stylization. In this experiment, we do not use style descriptions in the prompt to solely focus on demonstrating the effectiveness of the image-based style descriptor.
> >
> > ### Reference
> >
> > [1] Wu, Qiucheng, Yujian Liu, Handong Zhao, Ajinkya Kale, Trung Bui, Tong Yu, Zhe Lin, Yang Zhang, and Shiyu Chang. "Uncovering the disentanglement capability in text-to-image diffusion models." In Proceedings of the IEEE/CVF conference on computer vision and pattern recognition, pp. 1900-1910. 2023.
> >
> > [2] Tarrés, Gemma Canet, Dan Ruta, Tu Bui, and John Collomosse. "Parasol: Parametric style control for diffusion image synthesis." In Proceedings of the IEEE/CVF Conference on Computer Vision and Pattern Recognition, pp. 2432-2442. 2024.
> >
> > [3] Ruta, Dan, Gemma Canet Tarrés, Andrew Gilbert, Eli Shechtman, Nicholas Kolkin, and John Collomosse. "Diff-nst: Diffusion interleaving for deformable neural style transfer." arXiv preprint arXiv:2307.04157 (2023).
> >
> > [4] Wang, Haofan, Matteo Spinelli, Qixun Wang, Xu Bai, Zekui Qin, and Anthony Chen. "Instantstyle: Free lunch towards style-preserving in text-to-image generation." arXiv preprint arXiv:2404.02733 (2024).
> >
> > [5] Xu, Xingyu, and Yuejie Chi. "Provably robust score-based diffusion posterior sampling for plug-and-play image reconstruction." arXiv preprint arXiv:2403.17042 (2024).

---

> > > ### Author Response · Authors · 2024-11-22
> > > **Requesting Feedback on the Rebuttal**
> > >
> > > Dear Reviewer iCFZ,
> > >
> > > We hope the above clarifications and the additional experiments in the revised draft sufficiently address your concerns. If you are satisfied, we kindly request that you consider revising your score. We remain committed to addressing any remaining points you may have during the discussion phase.
> > >
> > > Best,
> > >
> > > Authors of Paper #11075

---

> > > > ### Comment · Reviewer_iCFz · 2024-11-24
> > > >
> > > > My review summarises this work as a technically sound paper but that does not fully contextualize its contribution in the literature or fully evidence its practicality.
> > > >
> > > > On the first point, i cited three prior works that reduce the novelty claims made over the fine grained controllability. Unfortunately these works have not been discussed in the revised paper, but simply a subsection added to the appendix repeating the rebuttal text.  Please properly integrate this response into Sec.1 / 2 the paper, to contextualise the novelty claims being made in order to address my first point.
> > > >
> > > > On the second point, Figure 18 has been added that shows gradual increase in style transfer strength for controllability. This partially addresses my concern as the figure shows the effect of increasing the weight of the effect.  However it doesn't show interpolation in the continuous style space.  Showing the latter would have better evidenced the value and practicality of the continuous representation.
> > > >
> > > > Since neither category of concern in my original review has been fully addressed by the revision, I will keep my score for now.

---

> ### Author Response · Authors · 2024-11-24
> **Discussion with Reviewer iCFz**
>
> Dear Reviewer iCFz,
>
> Thank you for your feedback on the rebuttal. Below, we address your remaining two points.
>
> **(Q1) On the first point, i cited three prior works that reduce the novelty claims made over the fine grained controllability. Unfortunately these works have not been discussed in the revised paper, but simply a subsection added to the appendix repeating the rebuttal text. Please properly integrate this response into Sec.1 / 2 the paper, to contextualise the novelty claims being made in order to address my first point.**
>
> (A1) We have now integrated our response into Sec 2 (L117-120 and L129-131) with further details deferred to Appendix B.7 due to space limitations.
>
> **(Q2) On the second point, Figure 18 has been added that shows gradual increase in style transfer strength for controllability. This partially addresses my concern as the figure shows the effect of increasing the weight of the effect. However it doesn't show interpolation in the continuous style space. Showing the latter would have better evidenced the value and practicality of the continuous representation.**
>
> (A2) We are glad that the newly added results in Figure 18 partially addressed your concerns. In response to the reviewer’s suggestion, we have now included Figure 19 (L1294-1295), which demonstrates interpolation in the continuous style space. We thank the reviewer for the great suggestion. Figure 19 shows an additional capability of our method to generate novel and unseen styles by continuously interpolating between reference style images.
>
>
>
> ### Concluding Remarks
>
> We hope the above clarifications and the additional experiments in the revised draft have successfully addressed all your concerns. If satisfied, we kindly request you to consider revising your score. We are happy to address any additional points during the discussion phase.
>
> Best,
>
> Authors of Paper #11075

---

> > ### Comment · Reviewer_iCFz · 2024-11-25
> >
> > Thank you for these further modifications.
> >
> > Regarding my first point, Section 2 has been updated to discuss the contribution in light of the prior related works.
> >
> > Regarding my second point, the new Figure 19 provides a convincing example of style interpolation although a fine-grained example e.g. between variants of a similar style would have been more convincing.
> >
> > Overall I consider my main concerns are addressed (or mostly addressed) now, and I will increase my score accordingly.

---

### Official Review · Reviewer_NtHN · 2024-11-05

**Soundness:** 4
**Presentation:** 3
**Contribution:** 4
**Rating:** 8
**Confidence:** 4

**Summary:**

This paper introduces reference-based modulation (RB-Modulation) for training-free personalization of diffusion models. The modulation builds on concepts from stochastic optimal control to modulate the drift field of reverse diffusion dynamics, incorporating desired attributes (e.g., style or content) via a terminal cost. Besides, the author also proposes Attention Feature Aggregation (AFA) module to decouple content and style in the cross-attention layers. The qualitative and quantitative results verify its effectiveness.

**Strengths:**

1. The idea is novel. The author provides the first training-free personalization framework using stochastic optimal control.
2. The author provides theoretical justifications connecting optimal control and reverse diffusion dynamics.
3. The qualitative and quantitative results are very promising for not only stylization, but also content-style composition. The author also conducts user study to further verify its superiority.

**Weaknesses:**

1. As the title said, the method aims for personalization. Therefore, I recommend the author add some comparison with classical personalization method, not just style transfer methods.
2. The section on ablation study in the paper is too brief. Considering that the AFA and SOC models are central to the article, I recommend that the authors include numerical comparisons and more qualitative comparisons.
3. For the content-style composition experiment, I recommend the author to provide more complicated ref content (dog, sloth, cat are simple cases, or you can change the color of the ref to see whether the proposed method has achieved better content-style composition) to verify its effectiveness.
4. For situations that require ControlNet to control the layout, can the method generalize well to these scenarios?

**Questions:**

Please refer to the weaknesses part.

---

> ### Author Response · Authors · 2024-11-18
> **Official Response by Authors**
>
> Dear Reviewer NtHN,
>
> Thank you for noting that our approach is the **first novel training-free personalization framework with theoretical justifications connecting optimal control and reverse diffusion dynamics.** We also appreciate your positive feedback on the **promising qualitative and quantitative results for stylization and content-style composition, supported by a user study verifying its effectiveness.** Below, we address your remaining questions.
>
> **(Q1) As the title said, the method aims for personalization. Therefore, I recommend the author add some comparison with classical personalization methods, not just style transfer methods.**
>
> (A1) In Figure 15, we have now added comparison with classical personalization methods (e.g., DreamBooth, IP-Adapter and DreamBooth+LoRA) and not just style transfer methods (e.g., StyleDrop and InstantStyle).
>
> **(Q2) The section on ablation study in the paper is too brief. Considering that the AFA and SOC models are central to the article, I recommend that the authors include numerical comparisons and more qualitative comparisons.**
>
> (A2) Due to space limitations, we deferred a detailed ablation study including numerical and additional qualitative comparisons to Appendix B.3 (e.g., Figure 8) and Appendix B.5 (e.g., Figure 10). We kindly invite you to review these sections and let us know if further results would be helpful. We would be glad to provide additional results if needed.
>
> **(Q3) For the content-style composition experiment, I recommend the author to provide more complicated ref content (dog, sloth, cat are simple cases, or you can change the color of the ref to see whether the proposed method has achieved better content-style composition) to verify its effectiveness.**
>
> (A3) We would like to note that, even for simpler cases (e.g., dog, sloth, cat) as shown in Figure 4, existing training-free content-style composition methods struggle to handle **complex prompts**. In contrast, our method achieves results comparable to a leading training-based method, ZipLoRA, with at least a 20X speedup (Table 4). We appreciate the reviewer’s suggestion to change the color of the content image, and we have now included Figure 22 to demonstrate our method’s robustness in capturing content-specific features independently of color.
>
> **(Q4) For situations that require ControlNet to control the layout, can the method generalize well to these scenarios?**
>
> (A4) For situations requiring ControlNet to control the layout, our proposed method can easily adapt, as shown in the newly added Figure 17. Since ControlNet enhances only the denoising network, we can still minimize the terminal cost associated with the expected terminal state, ensuring that SOC remains effective in practice. Similarly, the AFA module can generalize seamlessly by substituting the default attention processor with the AFA module, even when ControlNet is present.

---

### Official Review · Reviewer_ufZj · 2024-11-07

**Soundness:** 3
**Presentation:** 3
**Contribution:** 3
**Rating:** 10
**Confidence:** 5

**Summary:**

This paper borrows some concepts from optimal control and applies them to the diffusion models, which are training-free. The method is SOTA for the problems attempted on image stylization and composition.

**Strengths:**

+ The paper presents theoretical underpinnings from optimal control. It uses those ideas to solve the training-free stylization problems similar to Deep Image Prior, which is somehow not cited in the paper.

+ The work has several merits, especially in the problem of prompt-based image stylization in the training-free framework, though a pre-trained diffusion model is used for generation.

+ The title RB-Modulation is pretty weird and does not suit the paper's excellent contribution. This must be revised to put the paper in the correct research context.

+ The idea and theoretical contribution are quite good, and this work could lead to more interest in similar works.

**Weaknesses:**

- Title needs revision

- Missing references such as Deep Image Prior need to be cited.

**Questions:**

* Sorry to quote from your limitation. However, I am curious to know how feature descriptors can help this work.

---

> ### Author Response · Authors · 2024-11-18
> **Official Response by Authors**
>
> Dear Reviewer ufZj,
>
> Thank you for highlighting that **the paper has several merits, especially in training-free stylization**. We appreciate your encouraging feedback that **the idea and theoretical contributions are quite good, and this could lead to more interest in similar works**. Below, we address your remaining questions.
>
> **(Q1) Title needs revision. The title RB-Modulation is pretty weird and does not suit the paper's excellent contribution. This must be revised to put the paper in the correct research context.**
>
> (A1) We would appreciate any suggestions the reviewer may have for a revised title. A key contribution of this paper is the modulation of the standard reverse SDE’s drift field using a stochastic optimal controller and disentangled attention features from a reference image. If satisfactory, we propose the following alternatives: “RB-Modulation: Stochastic Control Meets Disentangled Attention for Personalization” or “RB-Modulation: Training-Free Personalization using Reference-Based Modulation”.
>
> **(Q2) Missing references such as Deep Image Prior need to be cited.**
>
> (A2) We thank the reviewer for pointing us to the related work, Deep Image Prior [1], which we have now cited in the revised version of our paper.
>
>
> **(Q3) Sorry to quote from your limitation. However, I am curious to know how feature descriptors can help this work.**
>
> (A3) In our optimal control framework, a feature descriptor is used within the terminal cost function. We employ a contrastive style descriptor (CSD) that is trained on 3,840 style tags [2]. However, some unique styles may still fall outside its scope. Improving the feature descriptor could enhance our work to capture these distinctive styles.
>
> Moreover, adopting an alternative feature descriptor could broaden downstream applications beyond personalization. For instance, a descriptor capable of capturing attributes like lighting, shadow, or haze could facilitate tasks such as relighting an image or removing shadows and haze, which are extremely important in biometric authentication platforms.
>
>
> ### Reference
>
> [1] Ulyanov, Dmitry, Andrea Vedaldi, and Victor Lempitsky. "Deep image prior." In Proceedings of the IEEE conference on computer vision and pattern recognition, pp. 9446-9454. 2018.
>
> [2] Somepalli, Gowthami, Anubhav Gupta, Kamal Gupta, Shramay Palta, Micah Goldblum, Jonas Geiping, Abhinav Shrivastava, and Tom Goldstein. "Measuring Style Similarity in Diffusion Models." arXiv preprint arXiv:2404.01292 (2024).

---

> > ### Comment · Reviewer_ufZj · 2024-11-23
> > **Increased the rating.**
> >
> > Having looked at the rebuttal, which answers all my concerns, I am increasing my rating. I would like the title to be - “RB-Modulation: Training-Free Personalization using Reference-Based Modulation” as suggested by the authors.

---

> > > ### Author Response · Authors · 2024-11-23
> > > **Thank you for increasing the score**
> > >
> > > Dear Reviewer ufZj,
> > >
> > > Thank you for your prompt response. We are glad to hear that the additional experiments in the revised draft addressed all your questions. We have now revised the title as discussed. We truly appreciate your thoughtful review and your decision to increase the score.
> > >
> > > Best,
> > >
> > > Authors of #11075

---

### Public Comment · ~Yuchen_Liang4 · 2024-11-28
**Missing Reference 1/2**

Dear Authers and Reviewers,

I think this article is just a special case of "FreeDom: Training-Free Energy-Guided Conditional Diffusion Model" published at ICCV 2023. FreeDom is a well-known  work in this field which was completed a year ago. Frankly, the only innovation I see in this article is the proximal algorithm. It's hard to believe that the authors didn't notice FreeDom's work when they researched the literature.

Best,

Yuchen

---

> ### Author Response · Authors · 2024-11-28
> **Relation with FreeDoM Paper**
>
> Dear Yuchen,
>
> Thank you for bringing FreeDoM [1] to our attention. We have now included a citation to this missing reference, highlighting **three technical differences between FreeDoM and our approach** (L1311-1321):
>
> 1. There are two algorithms proposed in FreeDoM. In their first Algorithm 1, the conditional guidance term $\nabla_{x_t} \log p(c|x_t)$ is approximated by the gradient of an energy function $\nabla_{x_t} \mathcal{E}(c,x_t)$. Our Algorithm 1 differs from this approach by replacing $\nabla_{x_t} \log p(c|x_t)$ with a controller guidance parameter $u$ that is **optimized** to minimize this approximation error. Furthermore, recent work [2] shows that such an optimization step, as in our Algorithm 1, is provably robust for posterior sampling, whereas Algorithm 1 in FreeDoM is provably biased. We note that either algorithm (ours or the one in FreeDoM) is hard to scale up to high dimensions because evaluating the gradient requires backpropagating through the score network, which is expensive for large-scale models.
>
> 2. In FreeDoM, Algorithm 2 is proposed to reduce the error in Algorithm 1 by integrating an efficient version of a time-travel strategy, where each denoised latent is noised back and forth to alleviate the poor guidance problem. Since this resampling process is expensive, we seek an alternative approach motivated by an optimal control framework. In our approach (Algorithm 2), we optimize the expected terminal state to satisfy the terminal constraint (e.g., align the style of the generated image with that of the given input), which is significantly different from FreeDoM. Our proximal approach allows us to avoid gradient computation through the denoising score network which is expensive for large-scale foundation models, such as SDXL or StableCascade. Indeed for StableCascade, it is not straightforward to even implement Algorithm 1 because StableCascade is a hierarchical diffusion model. We have discussed this at length in Section 4 (L216-234) and Appendix B.1.
>
> 3. Finally, we propose an Attention Feature Aggregation (AFA) module that effectively disentangles the content and style by leveraging a contrastive style descriptor (Section 4(b)). Our ablation study in Section 6.1, Appendices B.5 and B.6 demonstrate the importance of the AFA module in stylization and content-style disentanglement. We would like to note that FreeDoM relies on the standard attention processor, which suffers from content leakage as discussed in prior works [3,4].
>
>
> ### References
>
> [1] Yu, Jiwen, Yinhuai Wang, Chen Zhao, Bernard Ghanem, and Jian Zhang. "Freedom: Training-free energy-guided conditional diffusion model." In Proceedings of the IEEE/CVF International Conference on Computer Vision, pp. 23174-23184. 2023.
>
> [2] Xu, Xingyu, and Yuejie Chi. "Provably robust score-based diffusion posterior sampling for plug-and-play image reconstruction." arXiv preprint arXiv:2403.17042 (2024).
>
> [3] Hertz, Amir, Andrey Voynov, Shlomi Fruchter, and Daniel Cohen-Or. "Style aligned image generation via shared attention." In Proceedings of the IEEE/CVF Conference on Computer Vision and Pattern Recognition, pp. 4775-4785. 2024.
>
> [4] Wang, Haofan, Matteo Spinelli, Qixun Wang, Xu Bai, Zekui Qin, and Anthony Chen. "Instantstyle: Free lunch towards style-preserving in text-to-image generation." arXiv preprint arXiv:2404.02733 (2024).

---

> ### Public Comment · ~Yuchen_Liang4 · 2024-11-28
> **Discussion with Authors**
>
> Dear  Authers,
>
> Thank you for the authors' response. I acknowledge that Alg. 2 and AFA are valuable. Considering the high similarity between Alg. 1 and FreeDom, reviewers may have overestimated the contribution of this paper. Furthermore, FreeDom and its subsequent work, "DragonDiffusion: Enabling Drag-style Manipulation on Diffusion Models," are capable of performing a wide range of editing tasks. This paper only attempted stylization, which may appear somewhat limited.
>
> Best,
>
> Yuchen

---

> > ### Author Response · Authors · 2024-11-28
> > **Official Comment by Authors**
> >
> > We are responding to all the comments in one thread, please see above.

---

### Public Comment · ~Yuchen_Liang4 · 2024-11-28
**Miss  Reference 2/2**

Dear Authers and Reviewers,

I find that the proximal algorithm 2 proposed in this paper is also an existing method in the field of image processing.
It's a technique that's been used many times on other missions. In particular, "Manifold preserving guided diffusion" has tried this technique on FreeDom including stylization  task.

H. Chung, S. Lee, and J. C. Ye, “Decomposed diffusion sampler for accelerating large-scale inverse problems,” arXiv preprint arXiv:2303.05754, 2023.

Y. Zhu, K. Zhang, J. Liang, J. Cao, B. Wen, R. Timofte, and L. Van Gool, “Denoising diffusion models for plug-and-play image restoration,” in Proceedings of the IEEE/CVF Conference on Computer Vision and Pattern Recognition, 2023, pp. 1219–1229.

Chung H, Ye J C, Milanfar P, et al. Prompt-tuning latent diffusion models for inverse problems[J]. arXiv preprint arXiv:2310.01110, 2023.

Y. He, N. Murata, C.-H. Lai, Y. Takida, T. Uesaka, D. Kim, W.-H. Liao, Y. Mitsufuji, J. Z. Kolter, R. Salakhutdinov et al., “Manifold preserving guided diffusion,” in The Twelfth International Conference on Learning Representations, 2023.

Best,

Yuchen

---

> ### Author Response · Authors · 2024-11-28
> **Additional Discussion with Related Works (1/2)**
>
> **[Thank you for the authors' response. I acknowledge that Alg. 2 and AFA are valuable. I find that the proximal algorithm 2 proposed in this paper is also an existing method in the field of image processing. It's a technique that's been used many times on other missions. In particular, "Manifold preserving guided diffusion" has tried this technique on FreeDom including stylization task.]**
>
> Thank you for acknowledging (in comment 1/2) that Algorithm 2 and AFA are indeed valuable contributions. We respectfully disagree with the comment regarding overestimation of our contributions: in both theoretical development and the details of implementation, **FreeDoM significantly differs from our approach**. Our previous response clearly articulated **three major technical differences** between these two approaches. Beyond these differences in the algorithm, your comments are also ignoring the theoretical novelty, which are important as these are where the algorithmic differences originate.
>
> More broadly, with respect to the current literature, we believe that the contributions of this paper are significant given the fact that: (a) it provides a theoretical framework for analyzing diffusion models via stochastic control theory; (b) it identifies an important source of content leakage and presents two training-free modules (SOC and AFA) to mitigate  this leakage by disentangling the treatment of content and style via their respective attention heads; (c) large-scale experiments achieving a new state-of-the-art performance in stylization and content-style composition. To validate our contributions, we have compared with the current state-of-the-art InstantStyle [ArXiv’2024], other leading methods including StyleAligned [CVPR 2024 Oral], DreamBooth [CVPR 2023], and ZipLoRA [ECCV 2025].
>
> **Novelty of SOC:** We are not aware of prior works that view stylization and content preservation as the solution to a stochastic optimal control with terminal constraints. Our formulation provides a novel theoretical perspective and this is valuable because it allows the derivation of new algorithms from a principled perspective: the resulting algorithm differs in that the guidance function is an optimization problem with respect to the style constraints. **This constrained stochastic optimal control view is the key contribution** – once we have developed this view, the proximal algorithm (Algorithm 2) emerges naturally. The proximal method is a well known optimization technique and has been used in many fields including image processing. Indeed as reviewer ufZj commented, this also has connections to “training-free stylization problems similar to Deep Image Prior”, which effectively leads to a proximal method for inverse problems. Our use of the proximal method is unique in the sense that it uses disentangled representation derived from the style and content attention heads of a pre-trained ViT (see also detailed discussion toward the end of this response). Until now, all the prior works relied on the standard CLIP which suffers from content leakage.
>
> **Novelty of AFA:** (In response to the comment we received through email from the Openreview portal, which seems to have been deleted on the webpage: “[I also find that AFA has been proposed in DragonDiffusion and used to control global appearance similarity.]”) We believe that there may have been a misunderstanding regarding the attention mechanism proposed in AFA and the above reference DragonDiffusion. DragonDiffusion uses DDIM inversion to find the initial latent and construct a memory bank, which cannot be applied to hierarchical diffusion models, such as StableCascade. Further, at the time of writing our paper, the state-of-the-art method for stylization called InstantStyle already highlighted the issues of content leakage in such methods. We have compared our approach with this state-of-the-art method for stylization and other leading methods including StyleAligned [CVPR 2024 Oral], DreamBooth [CVPR 2023], and ZipLoRA [ECCV 2025].
>
> **[Furthermore, FreeDom and its subsequent work, "DragonDiffusion: Enabling Drag-style Manipulation on Diffusion Models," are capable of performing a wide range of editing tasks. This paper only attempted style transfer, which may appear somewhat limited if viewed as an incremental work.]**
>
> We have compared our approach with the current state-of-the-art methods in style transfer as discussed above. Beyond stylization experiments considered in these prior works, we also demonstrate superior performance in content-style composition. We provided empirical evidence showcasing capabilities of our method in the same experimental setting as these promising methods in the personalization community.

---

> > ### Author Response · Authors · 2024-11-28
> > **Additional Discussion with Related Works (2/2)**
> >
> > **[Relation to missing references]**
> >
> > The suggested references [1,2,3,4] study a different problem than personalization. Importantly, our proposed algorithm differs from these approaches as discussed below. Existing personalization methods [Section 2 in our paper] do not compare with these methods because these inverse problem solvers are time consuming for personalization tasks that demand fast inference.
> >
> > MPGD [1] takes a gradient step in the tangent space of the data manifold via projection with a (perfect) autoencoder. Our SOC module takes a gradient step in the pixel space (or latent space in LDMs) avoiding this extra projection via an autoencoder, **significantly improving the inference time**. Our framework allows us to do this because we personalize the pre-trained score function by a novel attention processor, the AFA module.
> >
> > DDS [2] assumes that the clean data manifold is represented by a Krylov subspace and the data distribution is uniform on this manifold. Under this assumption, DDS employs a proximal gradient descent algorithm to take a gradient descent in the tangent space. Again, this is different from our SOC module and unrelated to the AFA module.
> >
> > DiffPIR [3] splits the projection step into two subproblems: (1) Gaussian denoising (2) data proximal. Both the subproblems do not use text-conditioning and rely on the standard attention processor, which is known to suffer from content leakage (please refer to StyleAligned, InstantStyle).
> >
> > P2L [4] optimizes the null embedding to mitigate the suboptimal performance in other inverse problem solvers using a fixed null embedding in the text-to-image diffusion prior.  While effective, P2L takes around 20 min to process a single image, ours take just 40 sec (see our discussion in Table 4). Besides, our algorithm does not optimize the null embedding, instead it proposes an efficient attention processor, the AFA module, offering a 5-20X speedup compared to training/finetuning based methods.
> >
> > While several methods [1, 2, 3, 4], similar to ours have used a proximal sampler, their underlying generative model is not personalized. We believe this is the reason why these methods require a significant amount of time in satisfying the terminal constraints. Our paper takes the first step in personalizing the underlying generative model via a novel attention processor, which we believe would open an interesting line of future works for other applications.
> >
> >
> > ### Reference
> >
> > [1] Y. He, N. Murata, C.-H. Lai, Y. Takida, T. Uesaka, D. Kim, W.-H. Liao, Y. Mitsufuji, J. Z. Kolter, R. Salakhutdinov et al., “Manifold preserving guided diffusion,” in The Twelfth International Conference on Learning Representations, 2023.
> >
> > [2] Chung, Hyungjin, Suhyeon Lee, and Jong Chul Ye. "Decomposed Diffusion Sampler for Accelerating Large-Scale Inverse Problems." In The Twelfth International Conference on Learning Representations.
> >
> > [3] Y. Zhu, K. Zhang, J. Liang, J. Cao, B. Wen, R. Timofte, and L. Van Gool, “Denoising diffusion models for plug-and-play image restoration,” in Proceedings of the IEEE/CVF Conference on Computer Vision and Pattern Recognition, 2023, pp. 1219–1229.
> >
> > [4] Chung H, Ye J C, Milanfar P, et al. Prompt-tuning latent diffusion models for inverse problems[J]. arXiv preprint arXiv:2310.01110, 2023.

---

> ### Public Comment · ~Yuchen_Liang4 · 2024-11-29
> **Discussion with Authors**
>
> Personalization needs to perform transformations to source images (e.g., rotating the view, changing clothes, scenes, postures, etc.). Emphasizes facial and feature consistency. This is far beyond just stylization. If the article emphasizes personalization, it should have the function of other personalization works.

---

> > ### Author Response · Authors · 2024-11-29
> > **Final Response by Authors**
> >
> > Thanks for your comments and discussion. We continue to disagree about the contributions of our paper.

---

> > ### Comment · Reviewer_NtHN · 2024-11-29
> > **Discussion with Authors**
> >
> > I agree with Yuchen's point of view. I think using "personalization" in the title is not very appropriate, as the article mainly focuses on stylization.

---

> > > ### Comment · Reviewer_iCFz · 2024-11-29
> > >
> > > I notice that reviewer ufZj also considers the title of the paper is misaligned to the contribution.  I also agree on the paper having primarily a stylization focus, and had recommended the discussion of the additional stylization works that use a continuous parameterization in that spirit. There appears to be a unanimous consideration of the reviewers now re: a need to defocus on 'personalization' in the title.
> > >
> > > I do think this is a correctable and minor issue and having read the above arguments I still believe there is sufficient contribution to support acceptance of the paper so I will keep my borderline accept score.

---

> ### Comment · Reviewer_ufZj · 2024-11-29
> **Title Revision**
>
> I agree.
>
> I would like to keep the rating provided the title is revised something like below.
>
> RB-Modulation: Training-Free Stylization using Reference-Based Modulation
>
> Further, I request authors not to make tall claims like personalization in the text of the paper. Over punching things which are not justifiable does not augur well for the advancement of research.

---

> ### Author Response · Authors · 2024-11-29
> **Official Comment by Authors**
>
> We thank the reviewers and Yuchen Liang for engaging with us in the discussion. Following the suggestion, we will change the title to: “**RB-Modulation: Training-Free Stylization using Reference-Based Modulation**”, and update the text to avoid emphasizing personalization in the revised version.

---

### Meta-Review · Area_Chair_tzqs · 2024-12-18

**Metareview:**

This paper tackles the task of image Stylization using Reference-Based Modulation (RB-Modulation). The proposed approach is based on the proposed stochastic optimal controller, resulting in drift term ensuring high fidelity to reference image style and content in the text prompt. Cross-attention-based feature aggregation was proposed for decomposing the style and content from the reference image. The experimental results show the good control of the content and style composition in the generated image. This work received overall positive comments from the reviewers, with final scores of 10, 8, 6, 8. The proposed RB-Modulation approach is training-free, demonstrating sufficient novelty and effectiveness in controlling the content-style composition with diffusion models. It can be accepted based on these contributions and the consensus of acceptance recommendations from reviewers.

**Additional Comments On Reviewer Discussion:**

This paper has under extensive discussions in the post-rebuttal phase. The reviewers ufZj suggested the title revision and missing references. Reviewer NtHN also raised questions on the title with “personalization”, which is not fully compared with traditional personalization methods. Reviewer NtHN inquires on more justification on content-style composition. Reviewer iCFz suggested some related works, experiments on more nuanced control over style i.e. fine-grained variations of similar style, etc. Reviewer YTQn suggested more results on more widely adapted baseline models (e.g. SDXL), results with non-generated reference images, and questioned on content identity preservation. The reviewers are mostly satisfactory with the authors’ responses. The public comments on this submission included the concern on the relations of the proposed RB-Modulation method with FreeDom, DragonDiffusion. The authors clarified on the comparisons with these previous works. However, the final version of this work should contain these revisions considering reviewers’ comments and public comments.

---

### Decision · Program_Chairs · 2025-01-22

Accept (Oral)